# Surface-immobilized cross-linked cationic polyelectrolyte enables CO$_2$ reduction with metal cation-free acidic electrolyte

Hai-Gang Qin[1], Yun-Fan Du[1], Yi-Yang Bai[1], Fu-Zhi Li[1], Xian Yue[1], Hao Wang[1], Jian-Zhao Peng[1] & Jun Gu [1] ✉

Electrochemical CO$_2$ reduction in acidic electrolytes is a promising strategy to achieve high utilization efficiency of CO$_2$. Although alkali cations in acidic electrolytes play a vital role in suppressing hydrogen evolution and promoting CO$_2$ reduction, they also cause precipitation of bicarbonate on the gas diffusion electrode (GDE), flooding of electrolyte through the GDE, and drift of the electrolyte pH. In this work, we realize the electroreduction of CO$_2$ in a metal cation-free acidic electrolyte by covering the catalyst with cross-linked poly-diallyldimethylammonium chloride. This polyelectrolyte provides a high density of cationic sites immobilized on the surface of the catalyst, which suppresses the mass transport of H$^+$ and modulates the interfacial field strength. By adopting this strategy, the Faradaic efficiency (FE) of CO reaches 95 ± 3% with the Ag catalyst and the FE of formic acid reaches 76 ± 3% with the In catalyst in a 1.0 pH electrolyte in a flow cell. More importantly, with the metal cation-free acidic electrolyte the amount of electrolyte flooding through the GDE is decreased to 2.5 ± 0.6% of that with alkali cation-containing acidic electrolyte, and the FE of CO maintains above 80% over 36 h of operation at −200 mA·cm$^{-2}$.

Electrochemical reduction of CO$_2$ at ambient temperature is a promising technique to store renewable electricity, fix CO$_2$ and produce valuable fuels and chemicals[1,2]. One of the greatest challenges to make this technique industrially feasible is to prevent the formation of (bi) carbonate at the cathode[3,4]. Alkaline solution is not a practical electrolyte for CO$_2$ reduction due to the reaction between OH$^-$ and CO$_2$[5,6]. When carrying out CO$_2$ reduction in near neutral electrolyte such as KHCO$_3$ solution, partial CO$_2$ is consumed by OH$^-$ ions generated on the cathode to form HCO$_3^-$ or CO$_3^{2-}$ instead of being reduced. The as-formed HCO$_3^-$ or CO$_3^{2-}$ ions migrate to the anode and are acidified by H$^+$ generated on the anode, which gives rise to CO$_2$ crossover from cathode to anode[3,7,8]. This process results in low utilization efficiency of CO$_2$. Moreover, the high resistance of the near neutral electrolyte leads to low energy efficiency at high current density[9].

Recently, techniques of CO$_2$ reduction with acidic electrolyte were developed to circumvent the as-mentioned issues[9-21]. Alkali cations play an important role to suppress hydrogen evolution

reaction (HER) and promote CO$_2$ reduction in acidic condition. However, alkali cations induce bicarbonate precipitation on the cathode during CO$_2$ reduction in acidic electrolyte, which is still an issue that hinders the sustainability of CO$_2$ reduction[22,23]. Fig. 1a illustrates the formation process of bicarbonate precipitate on the cathode. In spite of the low pH of bulk electrolyte, CO$_2$ is able to convert to HCO$_3^-$ in the vicinity of the cathode with high local pH[12,13,24]. The as-formed HCO$_3^-$ is then combined with the high concentration of alkali cations accumulated at the local environment of the cathode, causing the precipitation of alkali metal bicarbonate. The bicarbonate precipitate can destroy the hydrophobicity of the gas diffusion electrode (GDE) and thus induce flooding, which can block the channel for the mass transport of CO$_2$[5,23,25-27]. The inset of Fig. 1a shows the energy dispersive spectroscopy (EDS) mapping of the cross-section of Ag/GDE after CO$_2$ reduction with H$_2$SO$_4$-K$_2$SO$_4$ electrolyte, which explicitly shows the permeating of K$^+$-containing electrolyte through the GDE. Besides, when alkali cation-containing

[1]Department of Chemistry, Southern University of Science and Technology, Shenzhen, Guangdong, China. ✉e-mail: guj6@sustech.edu.cn

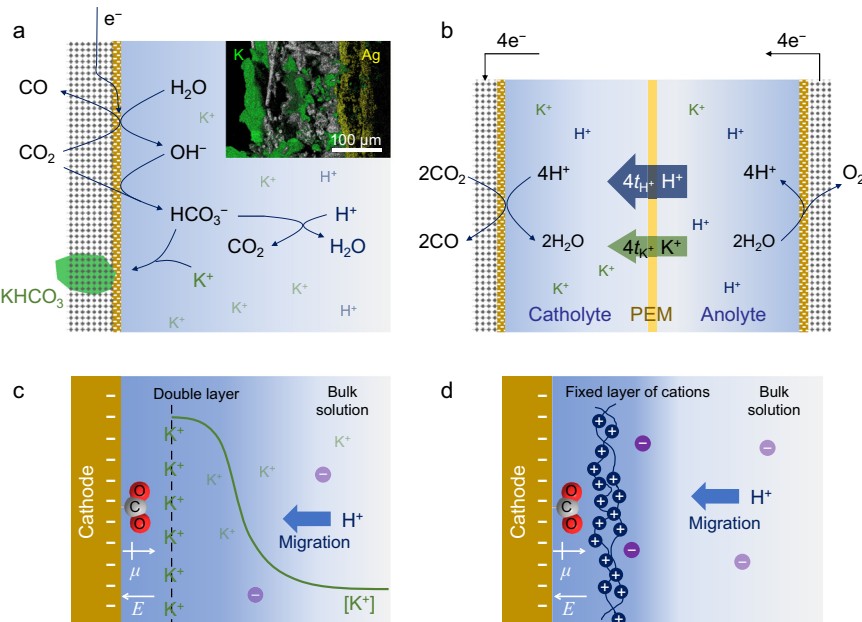

**Fig. 1 | Schematics of electrochemical CO₂ reduction with acidic electrolyte.** **a** Proposed mechanism of the formation of KHCO₃ precipitate on GDE with K⁺-containing acidic electrolyte. The inset shows the EDS mapping of the cross-section of Ag/GDE after CO₂ electroreduction experiment in 0.1 M H₂SO₄ + 0.4 M K₂SO₄. Green and yellow regions represent the distribution of K and Ag elements, respectively. **b** Migration of cations in the two-chamber cell with PEM. $t_{H+}$ and $t_{K+}$ represent the transference number of H⁺ and K⁺ in the PEM, respectively. **c** Effects of K⁺ ions on the adsorption of *CO₂ intermediate and the migration of H⁺. The dashed black line indicates the OHP. The green curve illustrates the concentration profile of K⁺. **d** Effects of immobilized layer of cations on the adsorption of *CO₂ intermediate and the migration of H⁺. $\mu$ and $E$ in **c**, **d** represent the dipole moment of adsorbed *CO₂ intermediate and the electric field in Stern layer, respectively.

acidic electrolyte is used in a two-chamber cell with proton exchange membrane (PEM), the pH values of catholyte and anolyte change in the long-term electrolysis. As illustrated in Fig. 1b, the mole H⁺ consumed on the cathode equals that generated on the anode. However, both H⁺ and alkali cations can traverse most PEMs, while the alkali cations cannot be consumed on the cathode. As a result, alkali cations accumulate in the catholyte and H⁺ ions accumulate in the anolyte, leading to the increase of pH of the catholyte and the decrease of pH of the anolyte. Therefore, a possible strategy to maintain the hydrophobicity of the GDE and the composition of the electrolyte during CO₂ reduction is to use metal cation-free acidic electrolyte.

Different mechanisms of cation effect in acidic electrolyte have been proposed. Hydrated cations at the outer Helmholtz plane (OHP) show distinct acidity compared with those in bulk solution, which is considered to tune the reactivity of HER and CO₂ reduction near the cathode[28,29]. More importantly, cations accumulate near the OHP modulate the distribution of electric field in the electric double layer. On one hand, cations screen the electric field generated from the cathode and thus retard the migration of H⁺[9,22]. On the other hand, cations enhance the electric field in Stern layer, which stabilizes the adsorbed polar intermediates of CO₂ reduction such as *CO₂ (Fig. 1c) and *OCCO[30–32]. Owing to these effects, H⁺ reduction is suppressed and CO₂ reduction is promoted. Moreover, it was reported that partially dehydrated alkali cations at the OHP can further stabilize the adsorbed polar intermediates via a short-range electrostatic interaction[29,33]. This effect was considered indispensable for enabling CO₂ adsorption and C-C coupling for the formation of C₂₊ products[29,33]. All the as-mentioned effects arise from cations accumulating near the cathode rather than those in the bulk electrolyte. Therefore, a layer of cations immobilized on the surface of the cathode may also enable CO₂ reduction in acidic condition without metal cations dissolved in the bulk electrolyte (Fig. 1d). Cationic polyelectrolytes are promising candidates for this task.

In this work, cross-linked poly-diallyldimethylammonium chloride (denoted as c-PDDA) was used as an immobilized cationic layer to cover the catalysts for CO₂ reduction. Aqueous solution of H₂SO₄ was used as the electrolyte. CO and formic acid were selectively generated with silver (Ag) and indium (In) nanoparticles (NPs) as the catalysts, respectively. The performance was stable over 36 hours and the pH values of both catholyte and anolyte kept constant. In sharp contrast, when K⁺-containing acidic electrolyte was used, the pH value of catholyte increased by 5.3 unit after electrolysis for 10 hours. Moreover, with metal cation-free acidic electrolyte, the amount of electrolyte flooding through the GDE was 1% of that with K⁺-containing acidic electrolyte, indicating that the hydrophobicity of the GDE was highly retained with metal cation-free electrolyte. This work provides the strategy to improve the sustainability of CO₂ electroreduction in acidic condition.

## Results and discussion

### CO₂ reduction in metal cation-free acidic electrolyte
In metal cation-containing acidic electrolyte, the electric field distribution is modulated by the accumulated cations in the electric double layer, which suppresses H⁺ reduction and promotes CO₂ reduction[9,22]. Inspired by this mechanism, we designed to immobilize a layer of cationic polyelectrolyte on the cathode to enable CO₂ reduction in metal cation-free acidic electrolyte. As shown in Fig. 2a, Ag NPs (99.99%, <100 nm, Macklin) showed negligible Faradaic efficiency (FE) of CO in a flow cell with 0.1 M H₂SO₄ as the electrolyte (black line). When Sustainion XA-9, a commercial cationic polyelectrolyte[34], was added with the catalyst, around 1% FE of CO was detected (orange line). The low FE of CO can be ascribed to the fact that the cation density of Sustainion XA-9 (2.06 mmol g⁻¹)[34] is not high enough to substantially affect the electric field distribution. Poly diallyldimethylammonium chloride (PDDA) possesses the cation density of 6.19 mmol·g⁻¹, 3 times that of Sustainion XA-9, which is the highest cation density we can find in a wide variety of polyelectrolytes. However, the PDDA decorated Ag

NPs shows unsatisfactory $CO_2$ reduction performance. The FE of CO was measured to be slightly higher than 2% at 15 min of electrolysis and dropped to below 1% after 30 min (blue line). The low performance of PDDA decorated Ag NPs can be attributed to the high solubility of PDDA in water. During the electrolysis, PDDA was washed away from the catalyst by the flowing catholyte and could not form a stable layer to cover the catalyst. Figure S1a, b compares the X-ray photoelectron spectroscopy (XPS) of PDDA decorated Ag catalyst before and after electrolysis. The peak of N $1s$ decreased substantially after electrolysis, indicating the loss of the layer of PDDA. [1]H-NMR spectrum of the electrolyte after electrolysis (Figure S2) indicates that about 90% of PDDA was washed into the electrolyte.

To prevent the layer of PDDA on the surface of catalyst from being washed away, we designed a cross-linked polyelectrolyte that can adhere firmly onto the catalyst, as illustrated in Fig. 2b. Figure S3 schematically shows the procedure of the preparation of c-PDDA decorated catalysts. First, we prepared the copolymer of diallyldimethylammonium chloride (DADMACl) and diallylmethylamine (DAMA). Figure S4 shows the characterizations of the copolymer. The N $1s$ XPS spectrum (Figure S4b) shows two peaks at 399.0 eV and 402.0 eV with the area ratio of 1:3.8, corresponding to the tertiary amine sites[35] and quaternary ammonium sites[36,37], respectively. 1,6-diiodohexane was chosen as the cross-linking agent to link the tertiary amine sites in two copolymer chains. To prepare the working electrode, the catalyst, the copolymer, 1,6-diiodohexane, ethylene glycol and ethanol were mixed as the ink and dropped onto the GDE. The GDE was then heated to facilitate the cross-linking reaction and evaporate the solvent. Figure S5 shows the high angle annular dark field-scanning transmission electron microscopy (HAADF-STEM) and EDS mapping images of Ag NPs coated with c-PDDA, in which N element distributes uniformly over Ag NPs. The N $1s$ XPS of this sample (Figure S1d) shows a single peak at 402.0 eV, indicating all the tertiary amine sites in the copolymer were converted to quaternary ammonium sites. The cation

density of c-PDDA was measured to be 5.80 mmol g[-1] and the charge density of c-PDDA immersed in water (denoted as $\rho_p$) was measured to be +296 C cm[-3] (corresponding to 3.07 M of cationic sites). The procedure of the measurement is shown in the Methods section. As shown by the magenta line in Fig. 2a, c-PDDA decorated Ag NPs presented the FE of CO up to 90% and kept above 80% during electrolysis for 36 h at −200 mA·cm[-2] in 0.1 M $H_2SO_4$. Figure S6 shows the electrode potential during the chronopotentiometry experiment. As presented in Figure S1c, d, the peak intensity of N $1s$ in the XPS spectra did not change obviously after electrolysis, confirming the c-PDDA layer was retained on the surface of Ag NPs. Figure S7 and S8 shows the infrared (IR) spectra and the electrochemical impedance spectra (EIS) of the working electrode before and after electrolysis, respectively. No substantial change in the spectra was observed, indicating both the polymer layer and the interface between Ag NPs and the polymer were stable during electrolysis. Inductively coupled plasma-mass spectroscopy (ICP-MS) analysis of the electrolyte after electrolysis showed that the concentrations of Na[+] and K[+] were 6.0×10[-6] M and 5.5×10[-6] M, respectively. To rule out the possibility that it was this trace amount of alkali cations rather than c-PDDA enabled $CO_2$ reduction in acidic electrolyte, 0.1 M of 18-crown-6 was added into 0.1 M $H_2SO_4$ to chelate alkali cations. The FEs of CO at −100 mA·cm[-2] in 0.1 M $H_2SO_4$ with and without 0.1 M of 18-crown-6 were 93% and 95%, respectively. The addition of 18-crown-6 did not obviously affect the FE of CO, indicating that the trace amount of alkali cations is not the origin of $CO_2$ reduction activity.

When K[+]-containing acidic solution (0.1 M $H_2SO_4$ + 0.4 M $K_2SO_4$) was used as the electrolyte, the FE of CO on bare Ag NPs also reached 90% initially. However, it dropped to below 80% after 2 hours and decreased to 43% after 10 hours (gray line in Fig. 2a). The poor stability of the electrocatalytic performance can be ascribed to the existence of K[+] in the acidic electrolyte. As shown by the magenta and orange lines in Fig. 2c, when 0.1 M $H_2SO_4$ was used as the electrolyte, the pH values

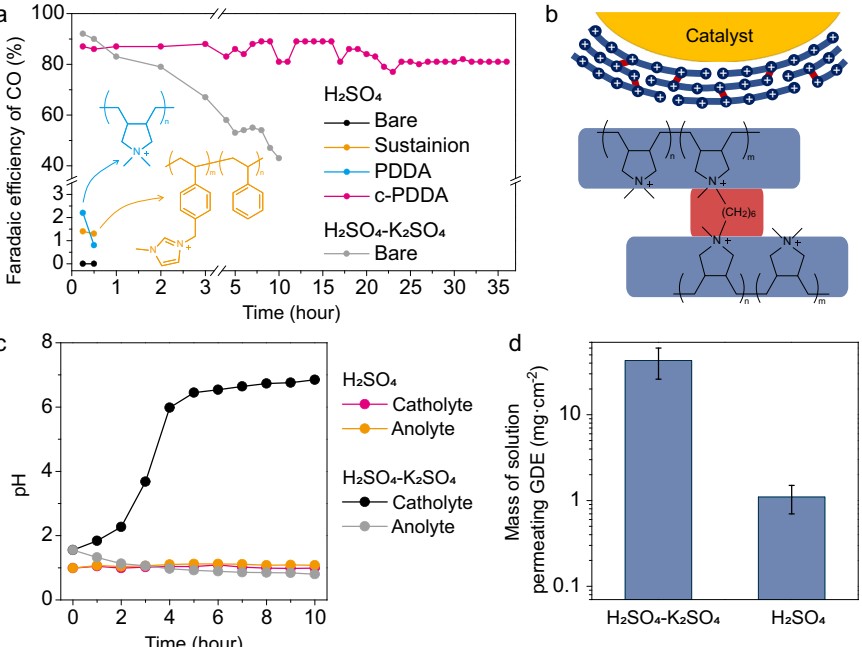

**Fig. 2 | $CO_2$ reduction with Ag NPs in acidic condition. a** FE of CO during electrolysis with constant current density of −200 mA·cm[-2]. Bare Ag NPs (black), Sustainion XA-9 decorated Ag NPs (orange), PDDA decorated Ag NPs (blue) and c-PDDA decorated Ag NPs (magenta) with 0.1 M $H_2SO_4$ as the flowing electrolyte, together with bare Ag NPs with 0.1 M $H_2SO_4$ + 0.4 M $K_2SO_4$ as the flowing electrolyte (gray). The insets show the structures of PDDA (blue) and Sustainion XA-9 (orange). **b** Schematic of c-PDDA decorated catalyst. The blue lines represent the polymer chains, and the short red lines represent the -$(CH_2)_6$- cross-linkers. **c** The pH values of the catholyte and anolyte during electrolysis with constant current density of −200 mA·cm[-2]. 0.1 M $H_2SO_4$ or 0.1 M $H_2SO_4$ + 0.4 M $K_2SO_4$ were used as the flowing electrolyte. **d** Mass of electrolyte permeating through the cathode after electrolysis with constant current density of −200 mA·cm[-2] for 10 h. Error bars are the standard deviations based on three individual measurements. Source data are provided as a Source Data file.

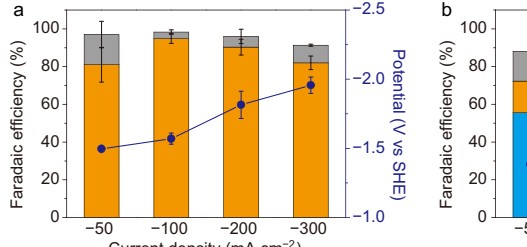
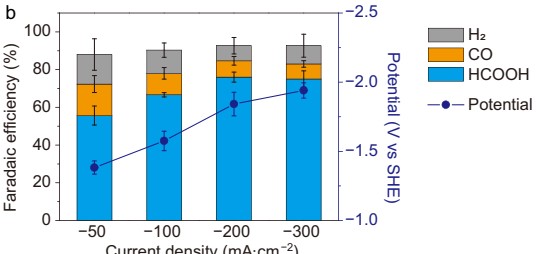

**Fig. 3 | CO₂ reduction performances of c-PDDA decorated catalysts in 0.1 M H₂SO₄. a** Ag NPs and **b** In NPs were used as the catalysts. Chronopotentiometry experiments were conducted. The FEs of H₂ (gray), CO (orange) and formic acid (blue), and the electrode potential (dark blue curves) are shown. Error bars are the standard deviations based on three individual measurements. Source data are provided as a Source Data file.

of both catholyte and anolyte kept around 1.0 in 10 h. Under this circumstance, $H^+$ is the only kind of ion that can migrate through the PEM. When 0.1 M $H_2SO_4$ + 0.4 M $K_2SO_4$ was used as the electrolyte, the pH of catholyte increased and the pH of anolyte decreased over time. Specifically, the pH values of the catholyte and the anolyte changed from 1.6 to 6.9 and 0.8 after 10 h, respectively (black and gray lines in Fig. 2c). Since the migrations of both $H^+$ and $K^+$ contribute to the current across the PEM (Nafion 211), the migration rate of $H^+$ through the PEM was lower than the generation rate of $H^+$ at the anode or the consuming rate of $H^+$ at the cathode (Fig. 1b). Consequently, the concentration of $H^+$ decreased and the concentration of $K^+$ increased in the catholyte. The pH of catholyte was convergent to about 7 because of the buffering effect of $CO_2$. The existence of $K^+$ in the electrolyte also led to the formation of $KHCO_3$ precipitate on the GDE, as indicated by X-ray diffraction (XRD) and scanning electron microscopy (SEM) characterizations (Figure S9). The increase of the pH and the concentration of $K^+$ in the catholyte facilitated the precipitation of $KHCO_3$. As shown in Figure S10, the contact angle of water on the side of gas diffusion layer of the working electrode after electrolysis in 0.1 M $H_2SO_4$ + 0.4 M $K_2SO_4$ was considerably smaller than that after electrolysis in 0.1 M $H_2SO_4$, indicating that the formation of $KHCO_3$ precipitate reduced the hydrophobicity of the GDE. The break-down of the hydrophobicity led to the electrolyte flooding and blocked the mass transport of $CO_2$. To quantify the electrolyte flooding, we measured the mass of electrolyte permeated through the GDE during 10 h of electrolysis. When 0.1 M $H_2SO_4$ + 0.4 M $K_2SO_4$ was used as the electrolyte and bare Ag NPs were used as the catalyst, 43 ± 17 mg of the electrolyte permeated through 1 cm² of GDE. When 0.1 M $H_2SO_4$ was used as the electrolyte and c-PDDA decorated Ag NPs were used as the catalyst, only 1.1 ± 0.4 mg of the electrolyte permeated through 1 cm² of GDE (Fig. 2d). The procedure of the measurement is shown in the Method section. It is noteworthy that the increase of the pH of the catholyte did not directly cause the decrease of the FE of CO on bare Ag NPs. As shown in Figure S11, the initial FEs of CO on bare Ag NPs in electrolytes with varied pH are all around 90%. The decrease of the FE of CO in 0.1 M $H_2SO_4$ + 0.4 M $K_2SO_4$ in Fig. 2a was a direct consequence of the formation of $KHCO_3$ precipitate.

Figure S12 further compares the FE of CO on bare Ag NPs in 0.1 M $H_2SO_4$ + $x$ M $K_2SO_4$ ($x$ = 0.04 ~ 0.4). As shown by our previous study[22], by decreasing the concentration of $K^+$, the stability was improved while the initial FE of CO decreased. 0.1 M $H_2SO_4$ + 0.1 M $K_2SO_4$ was an optimized composition of electrolyte that balance the FE of CO and stability[22]. In this electrolyte, the FE of CO was around 80% initially, lower than that in 0.1 M $H_2SO_4$ + 0.4 M $K_2SO_4$, while the decrease of FE of CO was slower. After 3 hours, the FE of CO became higher than that in 0.1 M $H_2SO_4$ + 0.4 M $K_2SO_4$ but significantly lower than that of c-PDDA decorated Ag NPs in 0.1 M $H_2SO_4$. The $CO_2$ reduction performance of c-PDDA decorated Ag NPs in 0.1 M $H_2SO_4$ + 0.4 M $K_2SO_4$ was also measured, as shown in Figure S12. The FE of CO was higher than bare Ag NPs in 0.1 M $H_2SO_4$ + 0.4 M $K_2SO_4$ in 10 hours, suggesting that the formation rate of $KHCO_3$ precipitate decreased. However, the

performance of c-PDDA decorated Ag NPs in 0.1 M $H_2SO_4$ + 0.4 M $K_2SO_4$ was less stable than in 0.1 M $H_2SO_4$, and $KHCO_3$ precipitate was still detectable after $CO_2$ reduction for 12 h (Figure S13), indicating that the c-PDDA layer can slow down but not prevent the formation of $KHCO_3$ precipitate.

Figure 3 shows the FEs of different products from $CO_2$ reduction in 0.1 M $H_2SO_4$ with c-PDDA decorated Ag NPs and In NPs as the catalysts. Figure S14 shows the TEM images and the XRD pattern of In NPs. CO and formic acid were selectively produced on Ag NPs and In NPs, respectively. For Ag NPs, the highest FE of CO was 95 ± 3%, achieved at −1.57 ± 0.04 V vs standard hydrogen electrode (SHE) at the current density of −100 mA·cm⁻². At the current density of 300 −mA·cm⁻², the FE of CO was 82 ± 4%, corresponding to a partial current density of CO of −246 ± 12 mA·cm⁻². When 1 M $H_2SO_4$ was used as the electrolyte, the highest FE of CO was 66%, achieved at the current density of −300 mA·cm⁻², corresponding to a partial current density of CO of −198 mA·cm⁻² (Figure S15). For In NPs, the highest FE of formic acid reached 76 ± 3%, at −1.84 ± 0.08 V vs SHE with the current density of −200 mA·cm⁻². At the current density of −300 mA·cm⁻², the FE of formic acid was 75 ± 4%, corresponding to a partial current density of formic acid of −225 ± 13 mA·cm⁻². In the electrolyte after $CO_2$ reduction experiments, Ag and In elements were not detectable by ICP-MS, indicating less than 0.2% of Ag or In was dissolved during electrolysis. Although the standard reduction potential ($\varphi°(In^{3+}/In)$ = −0.34 V vs SHE)[38] indicates metallic In can be oxidized by $H^+$ in a chemical reaction, In catalyst was stable in 0.1 M $H_2SO_4$ at the electrode potential for $CO_2$ reduction (<−1.3 V vs SHE). As shown in Figure S16, the selectivity of $CO_2$ reduction conducted on c-PDDA decorated catalysts in 0.1 M $H_2SO_4$ is higher than that on bare catalysts in 0.1 M $H_2SO_4$ + 0.1 M $H_2SO_4$ or 0.1 M $H_2SO_4$ + 0.04 M $H_2SO_4$ at varied current density. When undecorated Ag NPs and In NPs were used as the catalysts in 0.1 M $H_2SO_4$, $H_2$ was only reduction product (Figure S17). Figure S18 further compares the overall cell potential and each component of the potential loss at different current densities with 0.1 M $H_2SO_4$, 0.1 M $KHCO_3$ and 0.1 M KOH as the electrolyte. The potential loss is composed of ohmic loss, overpotentials of cathodic reaction and anodic reaction. The cell potential with 0.1 M $H_2SO_4$ is the lowest due to the lowest resistance of the electrolyte. The apparent overpotential of $CO_2$ reduction in 0.1 M KOH is the lowest[9,25], but KOH solution is not sustainable during electrolysis and is not a practical choice as the electrolyte for $CO_2$ reduction techniques. The cell potential with 0.1 M $KHCO_3$ is the largest due to the highest resistance of the electrolyte.

It was reported that immobilized quaternary ammonium cations on ionomer can enable $CO_2$ reduction on membrane electrode assembly (MEA) with pure water as the anolyte[39,40], implying that the quaternary ammonium cations have the ability to interact with *$CO_2$ species and promote $CO_2$ reduction in alkali cation-free condition. The interaction between quaternary ammonium cation and the adsorbed species weakens as the substituent groups on the N atom become bulkier[41]. The N atoms in PDDA bears two methyl groups, the smallest substituent group. Therefore, PDDA should show stronger interaction

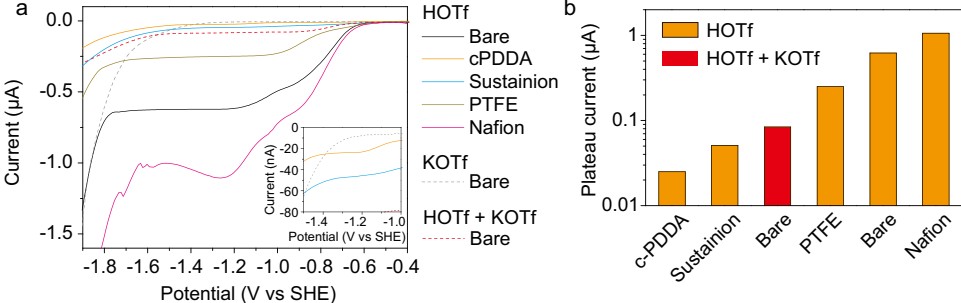

**Fig. 4 | Effect of polymer layer on the HER performance of Ag MDE. a** HER polarization curves of Ag MDEs in 10 mM HOTf (solid curves): Bare Ag MDE (black), Ag MDEs covered by c-PDDA (orange), Sustainion XA-9 (blue), PTFE (dark yellow) and Nafion D520 (magenta). The gray and red dashed curves show the HER polarization curve of bare Ag MDE in 10 mM KOTf and 10 mM HOTf + 10 mM KOTf, respectively. The inset shows the enlargement in the pink region. **b** Comparison of the plateau current in the HER polarization curves. The averaged current from −1.2 V to −1.5 V vs SHE was taken as the plateau current. Source data are provided as a Source Data file.

with *$CO_2$ species than quaternary ammonium cations with other substituent groups. In the alkali cation-containing electrolyte, the partially dehydrated alkali cation at OHP can bind to *$CO_2$ species, which is essential for triggering $CO_2$ reduction[29,33]. Since the quaternary ammonium cation cannot directly bind to *$CO_2$, the short-range interaction between *$CO_2$ and the quaternary ammonium cation should be weaker than that between *$CO_2$ and alkali cations. In addition, both alkali cations and quaternary ammonium cations can increase the electric field strength in Stern layer, which also stabilizes the polar *$CO_2$ intermediate, as discussed in the following section. Taking all the effects into account, $K^+$ should show more profound promotion effect on the kinetics of $CO_2$ reduction than c-PDDA, in accordance with our observation that the applied potential to reach the same partial current density of CO on bare Ag NPs in $K^+$-containing electrolyte (with $K^+$ concentration of 0.08 ~ 0.8 M) was more positive than on c-PDDA decorated Ag NPs in $K^+$-free electrolyte (Figure S19).

## Mechanism study on the effects of polymer layers

Our previous works indicate that alkali cations enable $CO_2$ reduction in acidic electrolyte through retarding the migration of $H^+$ and enhancing the electric field in Stern layer[9,22]. In this work, we used a layer of c-PDDA immobilized on the catalyst to take over the role of alkali cations in electrolyte. The effect of the cationic polyelectrolyte adlayers on $CO_2$ reduction in alkali cation-containing neutral electrolyte has been studied. In neutral electrolyte, the polyelectrolyte can modulate the concentration distribution of $HCO_3^-$, $CO_3^{2-}$ and $OH^-$ through Donnan exclusion and thus tune the local pH[41,42]. It was also reported that the polyelectrolyte layer can affect the local water content and $CO_2$ concentration[24,42]. In acidic electrolyte, the concentration distribution of $H^+$ can also be tuned by the polyelectrolyte through Donnan exclusion. More importantly, the polyelectrolyte layer may affect the rate of $H^+$ mass transport and hence determine the rate of $H_2$ evolution from $H^+$ reduction. The polyelectrolyte may also modulate the electric field in Stern layer and determines the rate of the electron transfer from cathode to $CO_2$. To understand these effects, we conducted experiments with Ag micro-disk electrode (MDE) covered by different polymers and simulated the experiments with generalized modified Poisson-Nernst-Planck (GMPNP) modeling[43,44].

The black curve in Fig. 4a shows the HER polarization curve of Ag MDE in 10 mM trifluoromethanesulfonic acid (HOTf). HOTf instead of $H_2SO_4$ was used as the electrolyte for the MDE experiments since HOTf dissociates completely in water, which helps to simplify the GMPNP modeling. The onset potential for HER is −0.6 V vs SHE, and a current plateau is observed at −1.2 V vs SHE. As the potential sweeps to more negative than −1.8 V vs SHE, the current increases again. Through the comparison with the HER polarization curve measured in 10 mM potassium trifluoromethanesulfonate (KOTf, gray dashed curve), we

can conclude that in 10 mM HOTf, the increase of current from −1.8 V vs SHE is originated from the reduction of water molecule, and $H^+$ reduction is the predominant contribution to the current between −0.6 V and −1.8 V vs SHE. The plateau region of the curve from −1.2 V to −1.8 V vs SHE is due to the limitation of the mass transport of $H^+$. As indicated by Fig. 3a, $CO_2$ reduction on Ag NPs was mainly observed at the potential of this plateau region. Therefore, $H^+$ reduction was the major competing reaction for $CO_2$ reduction, and suppressing $H^+$ reduction by retarding the mass transport of $H^+$ is a possible strategy to increase the FE of $CO_2$ reduction.

Furthermore, Fig. 4a compares the HER polarization curves of Ag MDEs covered by c-PDDA, Sustainion XA-9, polytetrafluoroethylene (PTFE) and Nafion D520, respectively, in 10 mM HOTf, and that of bare Ag MDE in 10 mM HOTf + 10 mM KOTf. Figure 4b compares the plateau current of $H^+$ reduction under different conditions. When the Ag MDE was covered by PTFE, a neutral polymer, the plateau was lower than that of bare Ag MDE, which can be ascribed to the lower diffusion coefficient of $H^+$ in the polymer layer than in aqueous solution. Ag MDE covered by Nafion D520, a polymer bearing anionic sites[45], showed higher plateau current than bare Ag MDE. Ag MDE covered by c-PDDA or Sustainion XA-9, polymers bearing cation sites, showed substantially lower plateau current than the Ag MDE covered by PTFE. Ag MDE covered by c-PDDA showed the lowest plateau current. These results indicate that the polymer layer bearing immobilized positive charge can suppress the mass transport of $H^+$, and the higher the positive charge density is, the lower the mass transport rate of $H^+$ is. It is noteworthy that the plateau current of bare Ag MDE in 10 mM HOTf + 10 mM KOTf is higher than that of cationic polymer decorated Ag MDE in 10 mM HOTf. Our previous study shows alkali cations can substantially suppress the migration rate of $H^+$ but the diffusion of $H^+$ cannot be significantly inhibited[22]. In metal cation-free solution, the migration rate of $H^+$ equals the diffusion rate of $H^+$ (see Supplementary Note 1 for the explanation). Therefore, once the migration of $H^+$ is suppressed by the cationic polymer layer, the diffusion of $H^+$ is suppressed simultaneously. As a consequence, the cationic polymer layer suppresses the mass transport of $H^+$ more substantially than dissolved alkali cations.

GMPNP simulation was then conducted to understand how the cationic polymer layer suppresses the mass transport of $H^+$. Figure 5a shows the model used for the simulation. Stern layer, the space between cathode and OHP, contains no charged species. A polymer layer is located outside the OHP with a charge density of the immobilized ionic sites of $\rho_p$. According to the loading of c-PDDA on Ag MDE in the experiments, the thickness of the polymer layer was set to 1 μm in the simulation. The polymer layer also contains movable $H^+$, $OH^-$ and $OTf^-$ ions. Outside the polymer layer, $H^+$, $OH^-$ and $OTf^-$ ions are the only charged species. The charge carried by the polymer layer modifies the

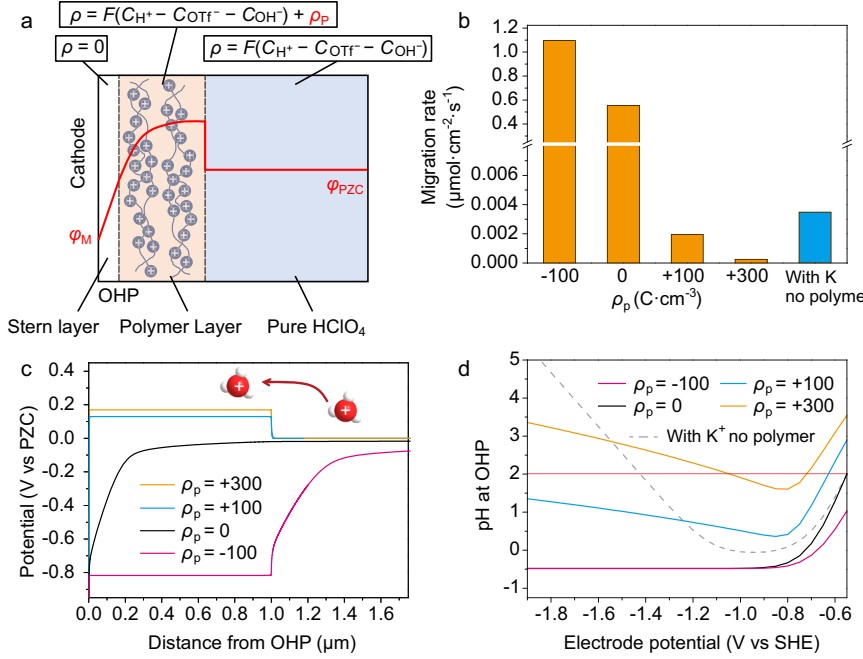

**Fig. 5 | Simulation of effects of polymer layer on the mass transport of H⁺.**
**a** Model used for the simulation. The red curve illustrates the potential profile. $\rho$ is the total charge density and $\rho_p$ is the charge density carried by the polymer layer. **b** Migration rate of H⁺ with the electrode potential of −1.8 V vs SHE at 2 μm from OHP. Orange bars: Ag electrode covered by polymer layers with different $\rho_p$ in 10 mM HOTf. Blue bar: Bare Ag electrode in 10 mM HOTf + 40 mM KOTf. **c** Potential profiles on Ag electrodes covered by polymer layers with different $\rho_p$ (unit: C·cm⁻³)

in 10 mM HOTf. The electrode potential is −1.8 V vs SHE. **d** Local pH at OHP of Ag electrodes at varied electrode potential. The solid curves represent Ag electrodes covered by polymer layers with different $\rho_p$ (unit: C·cm⁻³) in 10 mM HOTf. The gray dashed curve represents bare Ag electrode in 10 mM HOTf + 40 mM KOTf. The horizontal red line indicates the bulk pH. Source data are provided as a Source Data file.

electric field distribution according to Poisson equation:

$$\nabla^2 \varphi = -\frac{\rho}{\varepsilon_0 \varepsilon_r} \tag{1}$$

In this equation, $\varphi$ is the potential, $\rho$ is the total charge density of the immobilized ionic sites and movable ions, as shown in Fig. 5a, $\varepsilon_0$ is the permittivity of vacuum and $\varepsilon_r$ is the relative permittivity. Figure S20 shows the governing equations and boundary conditions used for the GMPNP simulation. Figure S21 shows the simulated polarization curves of H⁺ reduction with different $\rho_p$. The simulation reproduced the trend of the plateau current of H⁺ reduction in Fig. 4: The more positive $\rho_p$ is, the lower plateau current is. Figure 5b shows the migration rate of H⁺ outside the polymer layer at −1.8 V vs SHE. At this potential, H⁺ reduction is limited by the mass transport of H⁺. The migration rate of H⁺ decreases when $\rho_p$ increases, suggesting that cationic polymer layer can screen the electric field and suppress the migration of H⁺ while anionic polymer layer has the opposite effect. In addition, Fig. 5b shows that adding K⁺ ions into the acidic electrolyte can also suppress the migration of H⁺ on bare Ag electrode. In other word, both immobilized cations in the polymer layer and dissolved cations accumulated near the cathode can reduce the migration rate of H⁺, as illustrated in Fig. 1c, d. Moreover, Fig. 5c shows the simulated potential profiles with different $\rho_p$. When the polymer bears positive charge, the potential of the polymer layer is higher than the potential of bulk electrolyte, which is known as Donnan potential difference[46,47]. This layer with positive potential relative to bulk electrolyte acts as a barrier for the H⁺ mass transport from bulk electrolyte to the surface of cathode. The barrier becomes higher as $\rho_p$ of the polymer layer increases. Figure S22 shows the profiles of electric field strength with different $\rho_p$. Strong electric field exists at the interface between charged polymer layer and solution due to the potential step at the interface. For Ag electrode covered

by cationic polymer layer, the electric field is uniform within the polymer layer and solution. It is noteworthy that the electric field strength outside the polymer layer decreases as $\rho_p$ increases, in accordance with the migration rate of H⁺ shown in Fig. 5b.

The charge of the polymer layer also affects the local pH near the cathode. Figure S23 shows the pH profiles with different $\rho_p$ at −1.8 V vs SHE in 0.1 M HOTf. When the polymer layer is not charged, the pH at 10 nm to 1 μm from the OHP is higher than the bulk pH due to the concentration polarization. However, within 10 nm from the OHP, the pH is lower than the bulk pH since H⁺ ions are attracted electrostatically by the cathode. Figure 5d shows the pH at OHP at varied electrode potential. When $\rho_p$ is negative or zero (magenta and black curves), the negative shift of the electrode potential does not drive the increase of pH at OHP. When the polymer layer bears positive charge, the pH value in the polymer layer increases drastically (Figure S23), and the local pH at OHP increases as the electrode potential shifts negatively beyond -0.9 V vs SHE (blue and orange curve in Fig. 5d). When $\rho_p$ equals to +300 C·cm⁻³, the pH at OHP is higher than the bulk pH at the potential of the plateau region (<−1.2 V vs SHE). When K⁺ ions are contained in the acidic electrolyte and bare Ag electrode is used, pH at OHP increases as the electrode potential shifts negatively beyond −1.0 V vs SHE (gray dashed curve in Fig. 5d), in accordance with our previous reports[9,22]. Therefore, H⁺ reduction can drive local pH increase at OHP only when the cathode is covered by an immobilized cationic layer or the electrolyte contains inert cations. When $CO_2$ reduction is involved, OH⁻ anions generated from $CO_2$ reduction can neutralize H⁺ and lead to further increase of local pH[12]. Therefore, the local pH under $CO_2$ reduction condition should be higher than the value shown in Fig. 5d.

Besides retarding the migration of H⁺, the polymer layer also affects the electric field strength in Stern layer (denoted as $E_{Stern}$). The single-electron reduction of $CO_2$ ($CO_2 + e^- \rightarrow CO_2^-$) is regarded as the rate determining step (RDS) of $CO_2$ reduction to form CO in some

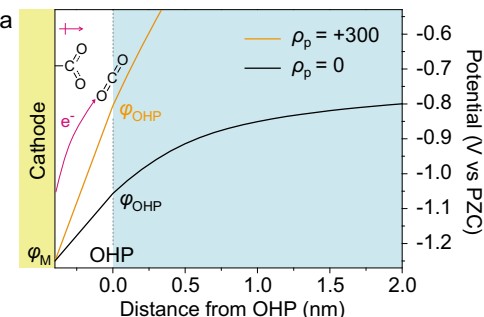

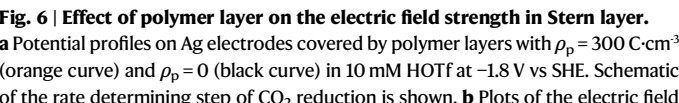

**Fig. 6 | Effect of polymer layer on the electric field strength in Stern layer.**
**a** Potential profiles on Ag electrodes covered by polymer layers with $\rho_p = 300$ C·cm⁻³ (orange curve) and $\rho_p = 0$ (black curve) in 10 mM HOTf at −1.8 V vs SHE. Schematic of the rate determining step of $CO_2$ reduction is shown. **b** Plots of the electric field

strength in Stern layer based on the electrode potential. Solid curves: Ag electrodes covered by polymer layers with different $\rho_p$ (unit: C·cm⁻³) in 10 mM HOTf. Gray dashed curve: Bare Ag electrode in 10 mM HOTf + 40 mM KOTf. Source data are provided as a Source Data file.

reports[48,49]. As illustrated in Fig. 6a, the potential difference between the cathode and OHP is the driving force of the electron transfer from the cathode to $CO_2$ molecules at OHP. It was also reported recently that chemisorption of $CO_2$ on the catalytic site is the RDS[50,51]. The *$CO_2$ species possesses the dipole moment towards the solution side and thus can be stabilized by the electric field in Stern layer. Therefore, no matter the reduction of $CO_2$ follows which mechanism, it can be accelerated by the increase of $E_{Stern}$[30–32,50].

The value $E_{Stern}$ was extracted from the result of GMPNP simulations. As shown in Fig. 6a, the black and the orange curves are the potential profiles on Ag electrode covered by polymer layers with $\rho_p$ of 0 and +300 C cm⁻³ at −1.8 V vs SHE, respectively. When $\rho_p$ equals to +300 C cm⁻³, $E_{Stern}$ (the slope of the curve in Stern layer) is higher, suggesting that the cationic polymer layer can promote $CO_2$ reduction on Ag electrode. Figure 5b compares the values of $E_{Stern}$ on Ag electrode covered by polymer layer with different $\rho_p$ at varied electrode potential. For Ag electrodes covered by neutral or anionic polymer layer ($\rho_p \leq 0$, black and magenta curves), $E_{Stern}$ first increases and then keeps constant as the electrode potential shifts negatively. Consequently, when the potential is more negative than −1.1 V vs SHE, $CO_2$ reduction cannot be accelerated by applying larger overpotential. For Ag electrode covered by cationic polymer layer ($\rho_p > 0$, blue and orange curves), $E_{Stern}$ increases continuously as the electrode potential shifts negatively. Moreover, $E_{Stern}$ is higher when $\rho_p$ is more positive. The dashed gray curve in Fig. 6b plots $E_{Stern}$ on bare Ag electrode in K⁺-containing acidic electrolyte. The promotion effect on $E_{Stern}$ induced by the polymer layer with $\rho_p$ of 300 C·cm⁻³ is comparable to the effect induced by the addition of K⁺ at −1.2 V to −1.9 V vs SHE. Considering that $CO_2$ reduction occurs on Ag catalyst in this potential range (Fig. 3a), a cationic polyelectrolyte with $\rho_p = 300$ C·cm⁻³ and K⁺ ions should show comparable promoting effect on $CO_2$ reduction.

We also simulated how the thickness of the polymer layer affects the migration of H⁺ and the value of $E_{Stern}$. As shown in Figure S24, the rate of H⁺ migration decreases as the thickness of the polymer layer increases, while the thickness of the polymer layer shows negligible effect on the value of $E_{Stern}$. Therefore, H⁺ reduction can be suppressed by increasing $\rho_p$ or increasing the thickness of the polymer layer, but $CO_2$ reduction can be promoted only by increasing $\rho_p$ of the polymer layer. If the polymer layer can be compressed under the electrostatic attraction generated from the cathode, $\rho_p$ at the cathode side should increase. As shown in Figure S25, the accumulation of cationic site to the cathode side leads to lower migration rate of H⁺ and higher $E_{Stern}$. Both effects result in improved selectivity of $CO_2$ reduction.

In summary, by decorating catalysts with c-PDDA, a cross-linked polyelectrolyte with high cation density, $CO_2$ reduction in metal cation-free acidic electrolyte was realized. CO and formic acid were produced with high Faradaic efficiency on Ag and In catalysts, respectively. Electrochemical measurements with MDE and GMPNP

simulations indicate that the cationic sites carried by the polymer layer play a similar role as alkali cations dissolved in acidic electrolyte on enabling $CO_2$ reduction, namely, suppressing the migration of H⁺ and enhancing the electric field in Stern layer. Conducting $CO_2$ reduction in metal cation-free acidic electrolyte helps to improve the sustainability of $CO_2$ reduction technique by maintaining the pH of electrolyte and inhibiting the flooding through GDE.

## Methods
### Chemicals
For the preparation of materials, Ag powder (99.99%, <100 nm, Macklin), InCl₃ (99.9%, Bide), NaBH₄ (98%, Aladdin), tetraethylene glycol (TEG, 99%, Aladdin), DADMACl (60% aqueous solution, Macklin), DAMA (99.5%, Bide), Ethylenediaminetetraacetic acid disodium salt (Na₂EDTA, 98%, Bide), (NH₄)₂S₂O₈ (99.99%, Macklin), NaOH (99%, Macklin), 1,6-diiodohexane (98%, Bide), ethylene glycol (99.5%, Xilong), Nafion D520 (5% in a mixture of lower aliphatic alcohols and water, DuPont), Sustainion XA-9 (alkaline ionomer 5% in ethanol, Dioxide Material) and PTFE (60 wt.% in water, 9dingchem) were used without further purification. For the preparation of electrolyte, H₂SO₄ (98%, ultrapure for trace metal analysis, Aladdin), K₂SO₄ (99.99%, Aladdin), HOTf (99%, Energy Chemical) and KOH (99.999%, Aladdin) were used. Pre-electrolysis (constant current electrolysis at −20 mA for 1 hour) was conducted for the HOTf solution before usage. Ultrapure water (18.2 MΩ cm) was used for all experiments.

### Materials synthesis
**In NPs**. 0.20 g of InCl₃ was dissolved in 30 mL of deionized water. 0.14 g of NaBH₄ was dissolved in 10 mL of TEG. The solution of NaBH₄ was dropped into the solution of InCl₃ under stirring and the mixture was further stirred at 25 °C for 2 h. In NPs were then separated by centrifugation and washed by ethanol for 3 times. Finally, In NPs were dried under vacuum at 60 °C for 12 hours.

**Copolymer of DADMACl and DAMA**. First, 6.9 mL of the aqueous solution of DADMACl (60 wt.%), 1.1 mL of DAMA and 0.20 mL of condensed H₂SO₄ were mixed to form a uniform solution. Next, 20 mg of (NH₄)₂S₂O₈ and 20 mg of Na₂EDTA were added into this solution. This solution was stirred at 60 °C in Ar atmosphere for 4 hours. Then, 0.3 g of NaOH was added to neutralize the solution. White precipitate was then formed by adding 10 mL of acetone. The precipitate was separated by filtration and washed by ethanol for 3 times. Finally, the precipitate was dried under vacuum to give the copolymer of DADMACl and DAMA.

### Electrode preparation
**c-PDDA decorated catalysts on GDE**. First, 0.5 mL aqueous solution containing 2.6 mg of the copolymer of DADMACl and DAMA, 10 mg of

the catalyst powder (Ag NPs or In NPs) and 0.5 mL of ethylene glycol were mixed and sonicated to form a uniform ink. Then, 20 µL of 1,6-diiodohexane was dissolved into 0.5 mL of ethanol and this solution was added into the ink and sonicated. The ink was loaded onto the GDE (Sigracet 29BC, 1×2 cm$^2$) on a heat plate at 170 °C with drop-casting method. The loading of the catalyst was 5 mg·cm$^{-2}$. The solvent was evaporated after 30 minutes. The electrode was then immersed in 10 mL of ethanol solution containing 40 µL of 1,6-diiodohexane and heat at 70 °C for 12 hours. Finally, the electrode was rinsed with ethanol and dried under vacuum.

**PDDA and Sustainion XA-9 decorated catalysts on GDE**. To prepare the ink of PDDA decorated catalyst, 1 mL of aqueous solution containing 3 mg of PDDA, 1 mL of ethanol and 10 mg of catalyst was mixed. To prepare the ink of Sustainion XA-9 decorated catalyst, 1 mL of ethanol solution containing 3 mg of Sustainion XA-9, 1 mL of deionized water and 10 mg of catalyst was mixed. The ink was sonicated for 1 hour and dropped onto the GDE (1 × 2 cm$^2$) on a heat plate at 120 °C. The loading of the catalyst was 5 mg cm$^{-2}$.

**Polymer covered Ag MDE**. Polycrystalline Ag MDE with the diameter of 25 µm was used. The MDE is embedded in a glass mantle with the diameter of 4 mm. To prepare Ag MDE covered by c-PDDA, the aqueous solution of the copolymer of DADMACl and DAMA was dilute to 3 mg mL$^{-1}$, and 2 µL of this solution was dropped onto the Ag MDE to cover the Ag surface and the mantle. The MDE was dried under infrared irradiation. Then, 40 µL of 1,6-diiodohexane was dissolved in 10 mL of ethanol, and the MDE was immersed in this solution at 70 °C for 12 h. Finally, the MDE was rinsed with ethanol and dried for further electrochemical measurement. To prepare Ag MDE covered by Nafion D520, Sustainion XA-9 or PTFE, the polymer dispersion was diluted by ethanol to 3 mg mL$^{-1}$, and 2 µL of this solution was dropped onto the Ag MDE to cover the Ag surface and the mantle. The MDE was dried for further electrochemical measurement.

## Characterizations

The XPS characterizations were performed on an ULVAC-PHI 5000 VersaProbe III XPS system using monochromatic Al K$_\alpha$ radiation (1486.6 eV). HAADF-STEM and EDS mapping images were collected on an FEI Tecnai F30 operated at 300 kV. XRD patterns were collected on a Rigaku SmartLab X-ray powder diffractometer with Cu K$_\alpha$ radiation ($\lambda$ = 1.5406 Å, 45 kV, and 200 mA). SEM characterizations were performed on ZEISS Merlin SEM with the EDAX Octane Pro energy dispersive X-ray spectroscopy system. ICP-MS measurements were conducted on Agilent 7700X ICP-MS. $^1$H-nuclear magnetic resonance ($^1$H-NMR) spectra were measured on AVANCE III 400 MHz. Element analysis was conducted on Elementar Vario EL cube. Molecular weight distribution of the copolymer was measured by Agilent gel permeation chromatography (GPC) 50 with water as the mobile phase.

**Measurements of charge density of c-PDDA**. A glass slide with known mass, thickness and area was used as a substrate. The layer of c-PDDA was deposited onto this glass slide with the same method of the preparation of the working electrode, except that the glass slide was used instead of the GDE and the catalyst was not added. Then, the mass and thickness of the slide after polymer deposition were measured to obtain the volume and the mass of the c-PDDA layer in dry form. Next, the slide was immersed in water for 30 minutes. The four sides of this slide were blocked by other glass slides to prevent the film of c-PDDA from swelling out of the slide. Then, the average thickness of the slide was measured by screw caliper to calculate the volume of the c-PDDA layer in wet form. The weight content of nitrogen in the dry form of c-PDDA was also measured through element analysis. The XPS spectrum (Figure S1d) indicates all N element in c-PDDA was in quaternary ammonium form. Thus, the cation density (unit: mmol·g$^{-1}$) of the dry

form of c-PDDA was calculated. From the mass and cation density of the dry form and the volume of the wet form, we calculated the charge density (unit: C·cm$^{-3}$) of the wet form of c-PDDA.

## Electrochemical measurements

**CO$_2$ reduction in flow cell**. All electrochemical experiments were conducted at 25 °C on an IVIUM potentiostat (Vertex.20 V.EIS). A three-chamber flow cell was used as the electrolyzer. The GDE as the working electrode and a Ag/AgCl/saturated KCl electrode as the reference electrode were in one chamber. All potential values were converted to SHE scale according to: $\varphi$(vs SHE) = $\varphi$(vs Ag/AgCl/saturated KCl) + 0.197 V. An IrO$_2$-decorated Ti foil as the counter electrode was in the other chamber. These two chambers were separated by a Nafion-211 membrane. 0.1 M H$_2$SO$_4$ was used as the flowing electrolyte. The catholyte and the anolyte were circulated separately by two peristaltic pumps. The volumes of catholyte and anolyte were both 30 mL and the flow rates were both 10 mL·min$^{-1}$. CO$_2$ was fed through a gas chamber behind the GDE. The flow rate was fixed at 30 standard cubic centimeter per minute (sccm) by a mass flow controller. Electrolysis was conducted with chronopotentiometry method. Resistance of the electrolyte was measured by impedance spectroscopy at −1.0 V vs SHE for all electrolytes (Table S1), and the real electrode potential ($\varphi_{real}$) was manually calculated with full $iR$ compensation after electrolysis according to $\varphi_{real} = \varphi_{measured} - iR$. The average Averaged potentials were then reported. The gas phase products (H$_2$ and CO) were quantified by an online gas chromatography (GC9790Plus, FULI INSTRUMENTS). The FEs of gas phase products were calculated based on the flow rate of the outlet gas from the flow cell measured by a soap film flowmeter. The liquid phase product (formic acid) was quantified by $^1$H-NMR with dimethyl sulfoxide (DMSO) as the inner standard. The pH values of catholyte and anolyte were measured by a pH meter. To measure the mass of electrolyte permeating through the GDE, a piece of filtrate paper and a piece of cotton wool were put behind the GDE (in the gas chamber) to absorb the solution passing through the GDE, and the change of the total mass of the filtrate paper and the cotton wool before and after electrolysis was weighed. The measurements of mass of electrolyte permeating the GDE (Fig. 2d) and the CO$_2$ reduction performances of c-PDDA decorated Ag NPs and In NPs (FEs and electrode potentials shown in Fig. 3a, b) were replicated with three electrodes prepared separately, respectively. The standard deviations were then calculated.

**HER on Ag MDE**. A Ag/AgCl/saturated KCl electrode was used as the reference electrode. At Pt wire was used as the counter electrode. The measurement was conducted in a single-chamber cell. 10 mM HOTf solution was used as the electrolyte. HER polarization curves were collected with linear sweep voltammetry (LSV) method with the sweeping rate of 100 mV s$^{-1}$. The $iR$ compensation was not conducted due to the small current.

## GMPNP simulation

GMPNP modeling was conducted to simulate H$^+$ reduction reaction in HOTf solution on Ag electrode covered by immobilized layer of polyelectrolyte. Figure S20 shows the geometry, governing equation and boundary conditions used for the simulation. Steady-state simulation was conducted. The conservation of mass gives:

$$\frac{\partial C_i}{\partial t} = - \nabla \cdot \boldsymbol{J}_i + R_i = 0 \tag{2}$$

In this equation, $\boldsymbol{J}_i$ is the flux of species $i$ ($i$ = H$^+$, OTf$^-$ and OH$^-$). $R_i$ is the rate of homogeneous reaction in which species $i$ is involved. OTf$^-$ does not participate in any homogeneous reaction, while H$^+$ and OH$^-$ are involved in neutralization reaction and dissociation of water. $\boldsymbol{J}_i$ is

expressed as:

$$\mathbf{J}_i = -D_i C_i \nabla(\ln(\gamma_i C_i)) - \frac{D_i C_i z_i F}{RT}\nabla\varphi \qquad (3)$$

In this equation, $D_i$, $C_i$ and $z_i$ are the diffusion coefficient, concentration, and charge of species $i$, respectively. $\varphi$ is the potential, $F$ is the Faradaic constant, $R$ is the gas constant, $T$ is the temperature (298 K). $\gamma_i$ is the 'Langmuir type' activity coefficient. The first and second terms on the right side of Eq. 3 correspond to the diffusion rate and the migration rate. The Poisson equation (Eq. (1)) and the conservations of mass (Eq. (2)) were solved simultaneously by COMSOL (v.6.0) with a MUMPS solver with a nonlinear automatic Newton method. Detailed procedures are in the 'Simulation section' of Supplementary Information. All parameters and coefficients used in the simulation are listed in Table S2. Considering that the values of $D_i$ are affected by the structure of polymer and the value of $\varepsilon_r$ varies according to the local environment, GMPNP simulations with different values of $D_i$ and $\varepsilon_r$ were also conducted to check whether the variation of these parameters affects the conclusion of the simulation. The results are summarized in Supplementary Note 2 (GMPNP simulations with varied diffusion coefficients and relative permittivity) and Tables S3-5.

## Data availability
Source data are provided in this paper. More data that support the findings of this study can be found in the Supplementary Information. Raw data are also available from the corresponding author upon request. Source data are provided in this paper.

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

## Acknowledgements

This work was supported by the National Natural Science Foundation of China (no. 22272073 to J.G.), Shenzhen Science and Technology Program (no. JCYJ20210324104414039, no. JCYJ20220818100410023 and no. KCXST20221021111207017 to J. G.), Guangdong Grants (2021ZT09C064 to J.G.), Guangdong Basic and Applied Basic Research Foundation (no. 2021A1515110360 and no. 2022A1515011976 to J.G.) and the "Climbing Program" Special Funds (no. pdjh2023c10301 to Y.-Y.B.).

## Author contributions

J.G. supervised this research project. H.-G.Q. conducted electrochemical measurements and simulations. Y.-F.D. and J.-Z.P. synthesized the copolymer. Y.-Y.B. conducted the MDE experiments. F.-Z.L. and X.Y. conducted sample characterization. H.-G.Q. and H.W. synthesized the catalysts. H.-G.Q. and J.G. wrote the manuscript.

## Competing interests

Chinese patent application (no. 2023107001339) was filed by the Southern University of Science and Technology with J. G., H.-G.Q., and Y.-F.D. as inventors. This patent pertains to the synthesis of c-PDDA and the uses of c-PDDA decorated Ag and In nanoparticles for electrocatalytic applications. Y.-Y.B., F.-Z.L., X.Y., H.W., and J.-Z.P. declare no competing interests.
