## [Peer Review File · Nature Communications]

REVIEWER COMMENTS

Reviewer #1 (Remarks to the Author):

In this manuscript, the authors reported interesting results about electrochemical CO₂ reduction in alkali cation-free electrolytes on Ag and In GDEs covered by cross-linked polydiallyldimethylammonium chloride. This cationic polymer immobilized on the catalyst surface can suppress H⁺ mass transport and modulates the local electric field strength to promote CO₂R, which is supported by their MDE and GMPNP simulations. The result in this manuscript is consistent with the authors' previous work about the effect of K⁺ on electrochemical CO₂R. However, the authors' theory contradicts another proposed mechanism which claims the indispensable role of alkali cations in the electrochemical CO₂R process because alkali cations are involved in the rate-determining step of CO₂R due to the direct interaction with adsorbed CO₂ or CO₂R intermediates (e.g. Nature Catalysis, 4, 654–662 (2021); Nature Communications 13, 5482 (2022)).

This reviewer suggests that this work can be considered accepted only after the authors answer the following questions and concerns in their manuscript.

1. In the introduction, the authors didn't do a good job of providing a comprehensive and accurate explanation/introduction of published proposed theories of alkali cation effects on electrochemical CO₂R in the last few years. Instead, the authors only emphasized their own understanding of the cation effect from their previous work but didn't honestly discuss another reported mechanism that claims the indispensable role of alkali cations in the electrochemical CO₂R process because alkali cations are involved in the rate-determining step of the CO₂R due to the direct interaction with adsorbed CO₂ or CO₂R intermediates (Ref 29 in this manuscript and Nature Communications 13, 5482 (2022)). The authors should not circumvent the discussion of this theory which seems to contradict the result of this manuscript but is really relative to the topic of the study of cation or cationic species effects on CO₂R in this work. The authors have to add a detailed introduction of this theory and add more discussion about the difference between these two theories and results in the RESULT AND DISCUSSION section.

2. Figure 2a showed that c-PDDA decorated Ag GDE displayed good FE of CO and long stability for 35 hours in 0.1 M H₂SO₄ while bare Ag showed declined stability with high FE of CO only in the first few hours in 0.1 M H₂SO₄ + 0.4 M K₂SO₄. The authors attributed the poor stability of bare Ag in 0.1 M H₂SO₄ + 0.4 M K₂SO₄ to the KHCO₃ precipitate on the GDE. Did the authors try smaller K⁺ concentrations (e.g. 0.1 M, 0.04M K⁺)? How is the stability of catalytic performance of bare Ag at lower [K⁺]? Is there any KHCO₃ precipitate by using lower [K⁺]? These control experiments are important because they can help to compare the authors' strategy of using cationic polymer on Ag in cation-free electrolytes vs. the currently applied strategy of using bare electrodes in cation-containing electrolytes as long as the precipitation issue doesn't happen.

3. If the cationic polymer on Ag can suppress H⁺ mass transfer, it might be able to suppress K⁺ mass transfer. To experimentally test the authors' theory, did the authors try to conduct CO₂R on c-PDDA decorated Ag in 0.1 M H₂SO₄ + 0.4 M K₂SO₄? Can a precipitate issue occur in this K⁺-containing electrolyte in the presence of cationic polymer on Ag?

4. In Figure 3, the authors showed the CO₂R performance of c-PDDA decorated catalysts in 0.1 M H₂SO₄. The authors should compare this result with the electrolysis result of bare Ag/In catalysts in 0.1M H₂SO₄ with optimized low [K⁺]. As mentioned in comment 2, these control experiments help to compare the authors' strategy of using cationic polymer on Ag/In in cation-free electrolytes vs. the currently applied strategy of using bare electrodes in cation-containing electrolytes as long as the precipitation issue doesn't happen.

5. In Figure 4, the authors should add the control experiment of bare Ag MDE in 10mM HOTf + 10mM KOTf for comparison.

6. In this manuscript, the authors claimed that the electrolytes they used are alkali cation free. Did the authors conduct ICP-MS measurements of their "alkali cation free" electrolytes to rule out the cation contamination which can probably come from the cell/membrane/tubes, c-PDDA Cl material, and the acid even though the acid and electrolyte salt are trace metal pure? Therefore, ICP-MS results are required to demonstrate the electrolyte is alkali-cation-free.

Reviewer #2 (Remarks to the Author):

The authors present a study of CO₂ electrolysis in a gas diffusion electrode, presenting an acidic system with an immobilized polyelectrolyte layer as a substitute for free alkali cations. This alternative approach suppresses the HER and bicarbonate precipitation. Consequently, surface hydrophobicity is maintained and flooding is prevented. I find the approach certainly intriguing, the study seems scientifically sound and the manuscript is insightful and well-written. As such, I support the publication of this manuscript after the authors address the following minor comments.

The authors indicate around line 230 that the effect of a polyelectrolyte layer on CO₂ reduction has been studied. They then continue to write what they will study in the present article. It should be stated more clearly how this work complements previous work and possibly builds on it. Simply put, is the

current work innovative, or is it rather similar to earlier work, but then with different conditions? Moreover, where previous studies did consider similar conditions, did the results agree?

From Figure 5c, I understand that the polymer layer is 1 micrometer thick. However, I overlooked this fact in the main manuscript. It would be wise to state this more clearly.

Looking at the different charge densities considered, I understand the Donnan potentials seen in figure 5c, but the rest of the profiles (zero or varying electric fields inside or beyond the polymeric layer) are not completely clear to me.

The electric field in the Stern layer is extracted from the GMPNP simulations, which aim to account for steric effect, but still might not be so accurate directly at the interface.

In the first place, the local properties in the EDL can be strongly affected by electrostatic correlations and molecular orientations. The authors account for a (rather drastic) drop of 90% in diffusion coefficient but for example not the effect of the polymer charge or local hydronium concentration on the dielectric permittivity. Could the authors justify their choice? For reference:

Zhu et al. (ref 46 in the manuscript) assumed a drop of over 90% in permittivity in the Stern layer.

A correlation to relate the local permittivity to concentration (in the case of free ions) was used for example by Bohra et al. (ref 41), and in a recent article that builds on the work of Bohra et al.:

Butt et al. *Sustainable Energy and Fuels* 7, 144-154 (2023), <https://doi.org/10.1039/D2SE01262F> (notably, that article also uses a Frumkin kinetic model in line with the present manuscript)

A change in the local permittivity would strongly affect the calculated E_{stern} and corresponding potential drop. As the authors mention in line 333, this is the driving force for the CO₂RR.

Reviewer #3 (Remarks to the Author):

This work achieved a significant enhancement of product selectivity and stability for the CO₂ electroreduction in an acidic catholyte free from metal cations by coating the catalyst with cross-linked diallyldimethylammonium (PDDA)-based polymer. Through a combined experimental and theoretical investigation, the authors reported that the high density of positively charged functional groups of the PDDA could serve a similar role of the alkali cations in retarding the proton migration close to the catalyst surface and enhancing the electric field within the Stern layer, and thus promote the electrode activity and stability in reducing CO₂ in acidic environment.

The loss of CO₂ in (bi)carbonates and salt precipitation are critical challenges limiting the application of CO₂ electrolysis at a scale. CO₂ electrolysis in an acidic environment free from metal cations is promising route in addressing these challenges. Therefore, the findings from this work are timely and interesting. However, there are a few concerns listed below for the authors to consider.

Recent reports (e.g., *Nature Catalysis*, 2021, 4, 654-662) highlighted that the metal cations are essential in activating CO₂ reduction via stabilizing the CO₂- intermediate via a short-range electrostatic interaction. The results from this work indicated that the non-metal cationic groups could also activate CO₂ reduction, which can be an interesting alternative perspective to the current understanding. However, it remains unclear to me in the main text how CO₂ reduction could proceed within the polymer environment. If the cationic site behaves similarly to a metal cation that stabilizes the intermediate via a short-range interaction, will the steric hinderance be an issue for the polymer? It would be also good if the authors could experimentally examine the polymer properties before and after high-rate CO₂ electrolysis.

Following up the above question, the use of XPS results may not be sufficient to support the authors' claim that the PDDA was washed away in line 126 – 133. Can the PDDA be detected and quantified in the electrolyte after the test? This additional result could help further support the authors' explanation and rule out the potential chemical degradation of the PDDA under CO₂ reduction conditions.

In Figure 2c, why the pH of the H₂SO₄-K₂SO₄ catholyte increases so significantly while anolyte's pH remains stable across the test? The authors should elaborate more experimental details to justify this point. If the pH changed so drastically in the catholyte, I don't think it is a fair comparison of the CO Faradaic efficiencies particularly between H₂SO₄-K₂SO₄ and c-PDDA test shown in Figure 2a.

The authors stated in line 192 – 193 that the loss of the electrode hydrophobicity is a result of the salt precipitation. However, there is no solid evidence from this work to support this statement. In addition, the polymer coated on the catalyst seems to be more hydrophobic than the bare catalyst based on metallic nanoparticles. The limited electrode flooding can be also partially contributed by the hydrophobic polymer coating. The authors should address this point in the main text.

When experimentally examining the effect of the polymer adlayer on proton migration and interfacial electric field, the authors used HOTf as the supporting electrolyte, which is different from the reaction environment for CO₂ electroreduction. The authors should explain why these two sets of experiment are translatable and there is no potential impact from the trifluoromethanesulfonate anions. I also found the explanation in line 239 – 244 lacks solid supporting evidence.

In the model, is it reasonable to assume the charge density of the polymer is uniform across the polymer layer? If not, how will the cationic sites be distributed in the polymer layer, and will such distribution changes the conclusion from the modelling results? The authors should talk about the limitation of their models.

Figure 5d shows that the polymer with 300 positive charge $C\text{ cm}^{-3}$ did not show a local pH as high as the case with K^+ . Is a $pH = 3$ sufficient to limit the availability of protons for the competitive HER? When calculating the local pH in the model, did the authors consider the water content within the polymer? The polymer with more fixed charges is expected to have more water molecules, does the model capture this feature?

Minor comments:

The authors should include the error bars calculated from three repetitive experiments for their key results. It is necessary to help the field to understand the repeatability of the results.

The experimental section, I could not see whether or not the authors measured the outlet flow rate from the flow cell. It is important to measure the outlet flow rates to calculate the FEs and understand the carbon utilisation efficiency. Additionally, the authors mentioned in the experimental section that they controlled the gas by using a mass flow meter, which guess should be a mass flow controller. A meter can only measure the flow rate.

Reviewer #4 (Remarks to the Author):

CO₂ reduction in acidic condition is a good idea for avoiding the formation of carbonate during CO₂ reduction. The idea of using cross-linked poly(2,2'-diallyldimethylammonium chloride) in the system of free cation is very interesting and the results of stability and selectivity are very encouraging. This manuscript requires a major revision before publication on Nature Communication.

I have few questions and comments as below:

What is the potential of oxidation reaction in anodic compartment? And the author should show and discuss about the cell potential during electrolysis. It is important to see the advantage as well as disadvantage of the system for CO₂ reduction in acidic condition.

I would recommend the author showing the result of electrochemical impedance before and after electrolysis of CO₂ reduction. It is important to understand more about electrochemical properties of cathodic electrode during CO₂ reduction.

To Reviewer 1:

General comments: In this manuscript, the authors reported interesting results about electrochemical CO₂ reduction in alkali cation-free electrolytes on Ag and In GDEs covered by cross-linked polydiallyldimethylammonium chloride. This cationic polymer immobilized on the catalyst surface can suppress H⁺ mass transport and modulates the local electric field strength to promote CO₂R, which is supported by their MDE and GMPNP simulations. The result in this manuscript is consistent with the authors' previous work about the effect of K⁺ on electrochemical CO₂R. However, the authors' theory contradicts another proposed mechanism which claims the indispensable role of alkali cations in the electrochemical CO₂R process because alkali cations are involved in the rate-determining step of CO₂R due to the direct interaction with adsorbed CO₂ or CO₂R intermediates (e.g. Nature Catalysis, 4, 654–662 (2021); Nature Communications 13, 5482 (2022)).

This reviewer suggests that this work can be considered accepted only after the authors answer the following questions and concerns in their manuscript.

Response: We highly appreciate the reviewer's comments. The point-to-point response to the comments can be found below. The corresponding revisions in the **main text** and **Supplementary Information** are highlighted in **yellow**.

Comment 1: In the introduction, the authors didn't do a good job of providing a comprehensive and accurate explanation/introduction of published proposed theories of alkali cation effects on electrochemical CO₂R in the last few years. Instead, the authors only emphasized their own understanding of the cation effect from their previous work but didn't honestly discuss another reported mechanism that claims the indispensable role of alkali cations in the electrochemical CO₂R process because alkali cations are involved in the rate-determining step of the CO₂R due to the direct interaction with adsorbed CO₂ or CO₂R intermediates (Ref 29 in this manuscript and Nature Communications 13, 5482 (2022)). The authors should not circumvent the discussion of this theory which seems to contradict the result of this manuscript but is

really relative to the topic of the study of cation or cationic species effects on CO₂R in this work. The authors have to add a detailed introduction of this theory and add more discussion about the difference between these two theories and results in the RESULT AND DISCUSSION section.

Response: In the Introduction section, we added a detailed introduction of the understandings of cation effects reported in *Nat. Catal.* **2021**, *4*, 654–662 (ref. 29) and *Nat. Commun.* **2022**, *13*, 5482 (ref. 33): “Moreover, it was reported that partially dehydrated alkali cations at the OHP can further stabilize the adsorbed polar intermediates via a short-range electrostatic interaction.^{29,33} This effect was considered indispensable for enabling CO₂ adsorption and C-C coupling for the formation of C₂₊ products.^{29,33}”

The quaternary ammonium sites in c-PDDA lack this kind of interaction with the *CO₂ intermediate. Meanwhile, the recent works of Zhuang *et al.* realized CO₂ reduction on MEA with pure water as the anolyte. In their reports (ref 39: *Nat. Energy* **2022**, *7*, 835; ref 40: *Electrochim. Acta* **2023**, *458*, 142509), the catalyst was coated with an ionomer with quaternary ammonium as the immobilized cationic site, which enabled CO₂ reduction in a condition free of alkali cations. Therefore, we opine that the quaternary ammonium cations may also have the ability to interact with *CO₂ species and promote CO₂ reduction. Koshy *et al.* reported that the interaction between a functionalized imidazolium cation and an adsorbed bicarbonate species weakens as the substituent group on the imidazolium becomes bulkier (ref 41: *JACS* **2021**, *143*, 14712). The quaternary ammonium site on PDDA bears two methyl group, the smallest substituent group. Therefore, the quaternary ammonium site on PDDA should show stronger interaction with *CO₂ species than the quaternary ammonium site with other substituent groups. Since the quaternary ammonium site cannot directly bind to *CO₂, this kind of interaction should be weaker than that between alkali cation and *CO₂. In addition to the short-range interactions, both alkali cations and quaternary ammonium cations can increase the electric field strength in Stern layer, which stabilizes the polar *CO₂ species, as illustrated by Figure 6b. Taking all of these effects into account, c-PDDA should exert weaker promotion effect on CO₂ reduction than K⁺ cations. As

shown in Figure S19, the applied overpotential to reach the same partial current density of CO₂ reduction on c-PDDA decorated catalysts in 0.1 M H₂SO₄ is larger than that on bare catalysts in 0.1 M H₂SO₄ + 0.4 M K₂SO₄. Through our strategy, larger overpotential is the price paid for the improved stability. In our opinion, for a sustainable technique, better stability is more important than smaller overpotential.

In page 13, we added the comparison between the two strategies: “It was reported that immobilized quaternary ammonium cations on ionomer can enable CO₂ reduction on membrane electrode assembly (MEA) with pure water as the anolyte,^{39,40} implying that the quaternary ammonium cations have the ability to interact with *CO₂ species and promote CO₂ reduction in alkali cation-free condition. The interaction between quaternary ammonium cation and the adsorbed species weakens as the substituent groups on the N atom become bulkier.⁴¹ The N atoms in PDDA bears two methyl groups, the smallest substituent group. Therefore, PDDA should show stronger interaction with *CO₂ species than quaternary ammonium cations with other substituent groups. In the alkali cation-containing electrolyte, the partially dehydrated alkali cation at OHP can bind to *CO₂ species, which is essential for triggering CO₂ reduction.^{29,33} Since the quaternary ammonium cation cannot directly bind to *CO₂, the short-range interaction between *CO₂ and the quaternary ammonium cation should be weaker than that between *CO₂ and alkali cations. In addition, both alkali cations and quaternary ammonium cations can increase the electric field strength in Stern layer, which also stabilizes the polar *CO₂ intermediate, as discussed in the following section. Taking all the effects into account, K⁺ should show more profound promotion effect on the kinetics of CO₂ reduction than c-PDDA, in accordance with our observation that the applied potential to reach the same partial current density of CO on c-PDDA decorated Ag NPs in K⁺-free electrolyte was more negative than on bare Ag NPs in K⁺-containing electrolyte (Figure S19).”

Figure S19. Plots of partial current density of CO dependent on the potential of working electrode for c-PDDA decorated Ag NPs in 0.1 M H₂SO₄ and bare Ag NPs in 0.1 M H₂SO₄ + 0.4 M K₂SO₄.

Comment 2: Figure 2a showed that c-PDDA decorated Ag GDE displayed good FE of CO and long stability for 35 hours in 0.1 M H₂SO₄ while bare Ag showed declined stability with high FE of CO only in the first few hours in 0.1 M H₂SO₄ + 0.4 M K₂SO₄. The authors attributed the poor stability of bare Ag in 0.1 M H₂SO₄ + 0.4 M K₂SO₄ to the KHCO₃ precipitate on the GDE. Did the authors try smaller K⁺ concentrations (e.g. 0.1 M, 0.04M K⁺)? How is the stability of catalytic performance of bare Ag at lower [K⁺]? Is there any KHCO₃ precipitate by using lower [K⁺]? These control experiments are important because they can help to compare the authors' strategy of using cationic polymer on Ag in cation-free electrolytes vs. the currently applied strategy of using bare electrodes in cation-containing electrolytes as long as the precipitation issue doesn't happen.

Response: The initial FE of CO decreased as the concentration of K⁺ decreased. In our previous work (ref 22: *ACS Catal.* **2023**, *13*, 916), we found KHCO₃ precipitation issue still existed when 0.1 M H₂SO₄ + 0.05 M K₂SO₄ was used as the electrolyte, as illustrated by the XRD and EDS-mapping images below:

Figure 7 of *ACS Catal.* 2023, 13, 916: Detection of KHCO₃ precipitation in GDEs after CO₂RR in acidic solution containing K⁺. Chronopotentiometry tests of Ag catalyst on GDEs at $-200 \text{ mA} \cdot \text{cm}^{-2}$ were conducted with solutions containing 0.1 M of H₂SO₄ and 0.01~0.4 M of K₂SO₄ for 12 hours. (a) XRD patterns of GDEs before and after CO₂RR tests. The diffraction peaks of graphitic carbon, silver and KHCO₃ are labeled by red circles, blue squares and pink triangles, respectively. The pink and blue vertical lines at the bottom are the standard diffraction peaks of monoclinic KHCO₃ (JCPDS no. 12-0292) and face-centered-cubic Ag (JCPDS no. 04-0783), respectively. (b) Zoom-in of the XRD patterns between 28° and 35° in panel (a). The pink dashed vertical lines indicate the diffraction peaks of KHCO₃. (c) EDS mapping of the cross section of GDE after CO₂RR test with C_{M+} of 0.8 M. Red, yellow, blue and pink regions in the EDS mappings correspond to C, O, Ag and K, respectively. (d-g) The channel of K K-edge of EDS mappings of the cross sections of GDEs after CO₂RR tests with C_{M+} of 0.02~0.8 M. All scale bars represent 100 μm. (h) FE of CO at 15 minutes of CO₂RR tests. The error bars are standard deviation based on three individual GDEs.

0.1 M H₂SO₄ + 0.1 M K₂SO₄ is an optimized composition of electrolyte that balance the FE of CO and stability. In this revised manuscript, we again tested the FE of CO of **bare Ag NPs** in 0.1 M H₂SO₄ + *x* M K₂SO₄ (*x* = 0.04 ~ 0.4) and reproduced the result, as shown in Figure S12. In 0.1 M H₂SO₄ + 0.04 K₂SO₄ (magenta curve), the initial FE of CO was only around 30%. In this circumstance, improved stability compared with in 0.1 M H₂SO₄ + 0.4 K₂SO₄ is meaningless. In 0.1 M H₂SO₄ + 0.1 M K₂SO₄ (blue curve), the initial FE of CO was around 80%, lower than that in 0.1 M H₂SO₄ + 0.4 M K₂SO₄. At the meantime, the stability was improved, and the FE of CO became higher than in 0.1 M H₂SO₄ + 0.4 M K₂SO₄ (orange curve) after 3 hours, but still lower than that of c-PDDA/Ag in 0.1 M H₂SO₄ (black curve).

In page 11, we added: “Figure S12 further compares the FE of CO on bare Ag NPs in 0.1 M H₂SO₄ + *x* M K₂SO₄ (*x* = 0.04 ~ 0.4). As shown by our previous study,²² by decreasing the concentration of K⁺, the stability was improved while the initial FE of CO decreased. 0.1 M H₂SO₄ + 0.1 M K₂SO₄ was an optimized composition of electrolyte that balance the FE of CO and stability.²² In this electrolyte, the FE of CO was around 80% initially, lower than that in 0.1 M H₂SO₄ + 0.4 M K₂SO₄, while the decrease of FE of CO was slower. After 3 hours, the FE of CO became higher than that in 0.1 M H₂SO₄ + 0.4 M K₂SO₄ but significantly lower than that of c-PDDA decorated Ag NPs in 0.1 M H₂SO₄.”

Figure S12. The FE of CO during electrolysis with constant current density of -200 mA·cm⁻² on bare Ag NPs in 0.1 M H₂SO₄ + *x* M K₂SO₄ (*x* = 0.04 ~ 0.4) and on c-PDDA

decorated Ag NPs in 0.1 M H₂SO₄ or in 0.1 M H₂SO₄ + 0.4 M K₂SO₄.

Comment 3: If the cationic polymer on Ag can suppress H⁺ mass transfer, it might be able to suppress K⁺ mass transfer. To experimentally test the authors' theory, did the authors try to conduct CO₂R on c-PDDA decorated Ag in 0.1 M H₂SO₄ + 0.4 M K₂SO₄? Can a precipitate issue occur in this K⁺-containing electrolyte in the presence of cationic polymer on Ag?

Response: We measured the CO₂ reduction performance of c-PDDA decorated Ag NPs in 0.1 M H₂SO₄ + 0.4 M K₂SO₄ (grey curve in Figure S12). The FE of CO was higher than bare Ag NPs in 0.1 M H₂SO₄ + 0.4 M K₂SO₄ (orange curve) in 10 hours, suggesting the formation rate of KHCO₃ precipitate decreased. However, the performance of c-PDDA decorated Ag NPs in 0.1 M H₂SO₄ + 0.4 M K₂SO₄ was less stable than in 0.1 M H₂SO₄ (black curve), and KHCO₃ precipitate was still detectable after CO₂ reduction for 12 hours, as shown by the XRD and EDS mapping in Figure S13. This result indicates that the c-PDDA layer cannot 100% inhibit the mass transport of K⁺. Since K⁺ migrate to the cathode cannot be consumed like H⁺, K⁺ cations still accumulate during electrolysis, leading to the formation of KHCO₃ precipitate.

In page 11, we added: “The CO₂ reduction performance of c-PDDA decorated Ag NPs in 0.1 M H₂SO₄ + 0.4 M K₂SO₄ was also measured, as shown in Figure S12. The FE of CO was higher than bare Ag NPs in 0.1 M H₂SO₄ + 0.4 M K₂SO₄ in 10 hours, suggesting the formation rate of KHCO₃ precipitate decreased. However, the performance of c-PDDA decorated Ag NPs in 0.1 M H₂SO₄ + 0.4 M K₂SO₄ was less stable than in 0.1 M H₂SO₄, and KHCO₃ precipitate was still detectable after CO₂ reduction for 12 hours (Figure S13), indicating that the c-PDDA layer can slow down but not prevent the formation of KHCO₃ precipitate”.

Figure S13. Characterizations of GDE with c-PDDA decorated Ag NPs after CO₂ electroreduction with 0.1 M H₂SO₄ + 0.4 M K₂SO₄ as the electrolyte. (a) XRD patterns. Red circles, blue squares and pink triangles represent the diffraction peaks from GDE, Ag and KHCO₃, respectively. (b) SEM images and EDS mapping of the cross-sections of c-PDDA/Ag/GDE after electrolysis. Yellow and green regions represent Ag and K elements, respectively. K element was detected in the gas diffusion layer of the GDE after electrolysis.

Comment 4: In Figure 3, the authors showed the CO₂R performance of c-PDDA decorated catalysts in 0.1 M H₂SO₄. The authors should compare this result with the electrolysis result of bare Ag/In catalysts in 0.1M H₂SO₄ with optimized low [K⁺]. As mentioned in comment 2, these control experiments help to compare the authors' strategy of using cationic polymer on Ag/In in cation-free electrolytes vs. the currently applied strategy of using bare electrodes in cation-containing electrolytes as long as the precipitation issue doesn't happen.

Response: We measured the CO₂ reduction performances of bare Ag and In catalysts at varied current density in 0.1 M H₂SO₄ + 0.1 M K₂SO₄ and 0.1 M H₂SO₄ + 0.04 M K₂SO₄. As shown in Figure S16, the FEs of CO on c-PDDA decorated Ag and formic acid on c-PDDA decorated In in 0.1 M H₂SO₄ are higher than those on bare catalysts in 0.1 M H₂SO₄ + 0.1 M K₂SO₄ or 0.1 M H₂SO₄ + 0.04 M K₂SO₄.

In page 12, we added: “As shown in Figure S16, the selectivity of CO₂ reduction conducted on c-PDDA decorated catalysts in 0.1 M H₂SO₄ is higher than that on bare catalysts in 0.1 M H₂SO₄ + 0.1 M H₂SO₄ or 0.1 M H₂SO₄ + 0.04 M H₂SO₄ at varied current density.”

Figure S16. Comparison of FEs of (a) Ag NPs and (b) In NPs at varied current density. c-PDDA decorated catalysts in 0.1 M H₂SO₄ and bare catalysts in 0.1 M H₂SO₄ + 0.1 M K₂SO₄ and 0.1 M H₂SO₄ + 0.04 M K₂SO₄ are compared.

Comment 5: In Figure 4, the authors should add the control experiment of bare Ag MDE in 10mM HOTf + 10mM KOTf for comparison.

Response: We added the HER polarization curve of bare Ag MDE in 10mM HOTf + 10mM KOTf in Figure 4 (red dashed curve). The plateau current of H⁺ reduction in this

condition is lower than that of bare Ag MDE in 10 mM HOTf (black solid curve) but higher than that of cationic polymer decorated Ag MDE in 10 mM HOTf (blue and orange solid curves).

Figure 4. Effect of polymer layer on the HER performance of Ag MDE. (a) HER polarization curves of Ag MDEs in 10 mM HOTf (solid curves): Bare Ag MDE (black), Ag MDEs covered by c-PDDA (orange), Sustainion XA-9 (blue), PTFE (dark yellow) and Nafion-117 (magenta). The grey and red dashed curves show the HER polarization curve of bare Ag MDE in 10 mM KOTf and 10 mM HOTf + 10 mM KOTf, respectively. The inset shows the enlargement in the pink region. (b) Comparison of the plateau current in the HER polarization curves. The averaged current from -1.2 V to -1.5 V vs SHE was taken as the plateau current.

According to our previous report (ref. 22: *ACS Catal.* **2023**, *13*, 916), K^+ can significantly suppress the migration rate of H^+ . Meanwhile, K^+ affects the diffusion rate of H^+ but not significantly suppress the diffusion. When 10 mM HOTf is used as the electrolyte, H^+ and OTf^- are the only ionic specie in the electrolyte. The fluxes of these two kinds of ions can be expressed as:

$$J_{x,H^+} = -D_{H^+} \frac{dC_{H^+}}{dx} - \frac{D_{H^+} C_{H^+} F}{RT} \frac{d\phi}{dx} \quad (S21)$$

$$J_{x,OTf^-} = -D_{OTf^-} \frac{dC_{OTf^-}}{dx} + \frac{D_{OTf^-} C_{OTf^-} F}{RT} \frac{d\phi}{dx} \quad (S22)$$

The first and second terms on the right side of each equation correspond to the diffusion and migration terms, respectively. Considering the electroneutrality, $C_{H^+} = C_{OTf^-} = C$. Since OTf^- is not involved in the electrode reaction, its flux should be zero at steady

state. Therefore, we have:

$$\frac{dC}{dx} = \frac{CF}{RT} \frac{d\varphi}{dx} \quad (\text{S23})$$

Thus:

$$J_{x,H^+} = -2D_{H^+} \frac{dC}{dx} = -2 \frac{D_{H^+} CF}{RT} \frac{d\varphi}{dx} \quad (\text{S24})$$

Namely, the migration rate of H^+ equals the diffusion rate of H^+ . Therefore, once the migration of H^+ is suppressed by the cationic polymer layer, the diffusion of H^+ is suppressed simultaneously.

This effect can also be understood from Donnan equilibrium. C_{H^+} changes abruptly across the interface between the cationic polymer layer and the electrolyte solution. As shown in Figure S23, C_{H^+} in the cationic polymer layer is orders of magnitude lower than C_{H^+} in the solution, and C_{H^+} outside the cationic polymer layer is already quite close to the bulk C_{H^+} (0.01 M). Thus, the diffusion rate of H^+ is low due to the low gradient of C_{H^+} .

In summary, the cationic polymer layer suppresses the migration and diffusion of H^+ simultaneously, while the diffusion of H^+ was not substantially suppressed on bare catalyst in K^+ -containing solution. Therefore, the plateau current of H^+ reduction on cationic polymer decorated Ag MDE in 10 mM HOTf was lower than that on bare Ag MDE in 10 mM HOTf + 10 mM KOTf.

Figure S23. Simulated (a) pH and (b) C_{H^+} profiles at -1.8 V vs SHE. Solid curves: Ag electrode covered by polymer layer with different ρ_p (unit: $\text{C} \cdot \text{cm}^{-3}$) in 10 mM HOTf. Dashed grey curve: bare Ag electrode in 10 mM HOTf + 40 mM KOTf.

In page 15, we added: “It is noteworthy that the plateau current of bare Ag MDE in 10 mM HOTf + 10 mM KOTf is higher than that of cationic polymer decorated Ag MDE in 10 mM HOTf. Our previous study shows alkali cations can substantially suppress the migration rate of H^+ but the diffusion of H^+ cannot be significantly inhibited.²² In metal cation-free solution, the migration rate of H^+ equals the diffusion rate of H^+ (see Supplementary Note 1 for the explanation). Therefore, once the migration of H^+ is suppressed by the cationic polymer layer, the diffusion of H^+ is suppressed simultaneously. As a consequence, the cationic polymer layer suppresses the mass transport of H^+ more substantially than dissolved alkali cations.” In Supplementary Information, we added Supplementary Note 1: Effects of alkali cations and cationic polymer layer on the mass transport of H^+ .

Comment 6: In this manuscript, the authors claimed that the electrolytes they used are alkali cation free. Did the authors conduct ICP-MS measurements of their “alkali cation free” electrolytes to rule out the cation contamination which can probably come from the cell/membrane/tubes, c-PDDA Cl material, and the acid even though the acid and electrolyte salt are trace metal pure? Therefore, ICP-MS results are required to demonstrate the electrolyte is alkali-cation-free.

Response: We did ICP-MS analysis of the H₂SO₄ electrolyte **after electrolysis**. The concentration of Na⁺ and K⁺ were 6.0×10⁻⁶ M and 5.5×10⁻⁶ M, respectively. Li⁺ and Cs⁺ were not detectable. To rule out the possibility that this trace amount of alkali cations enable CO₂ reduction in acidic condition, we added 0.1 M of 18-crown-6 into the electrolyte to chelate alkali cations in the electrolyte and conducted CO₂ reduction electrolysis. The FEs of CO at -100 mA·cm⁻² in 0.1 M H₂SO₄ with and without 0.1 M of 18-crown-6 were 93% and 95%, respectively. The addition of 18-crown-6 did not obviously affect the FE of CO, indicating that the trace amount of alkali cations is not the origin of CO₂ reduction activity.

In page 9, we added: “Inductively coupled plasma-mass spectroscopy (ICP-MS) analysis of the electrolyte after electrolysis showed that the concentrations of Na⁺ and K⁺ were 6.0×10⁻⁶ M and 5.5×10⁻⁶ M, respectively. To rule out the possibility that it was this trace amount of alkali cations rather than c-PDDA enabled CO₂ reduction in acidic electrolyte, 0.1 M of 18-crown-6 was added into 0.1 M H₂SO₄ to chelate alkali cations. The FEs of CO at -100 mA·cm⁻² in 0.1 M H₂SO₄ with and without 0.1 M of 18-crown-6 were 93% and 95%, respectively. The addition of 18-crown-6 did not obviously affect the FE of CO, indicating that the trace amount of alkali cations is not the origin of CO₂ reduction activity.”

To Reviewer 2:

General comments: The authors present a study of CO₂ electrolysis in a gas diffusion electrode, presenting an acidic system with an immobilized polyelectrolyte layer as a substitute for free alkali cations. This alternative approach suppresses the HER and bicarbonate precipitation. Consequently, surface hydrophobicity is maintained and flooding is prevented. I find the approach certainly intriguing; the study seems scientifically sound and the manuscript is insightful and well-written. As such, I support the publication of this manuscript after the authors address the following minor comments.

Response: We highly appreciate the reviewer's comments. The point-to-point response to the comments can be found below. The corresponding revisions in the **main text** and **Supplementary Information** are highlighted in **yellow**.

Comment 1: The authors indicate around line 230 that the effect of a polyelectrolyte layer on CO₂ reduction has been studied. They then continue to write what they will study in the present article. It should be stated more clearly how this work complements previous work and possibly builds on it. Simply put, is the current work innovative, or is it rather similar to earlier work, but then with different conditions? Moreover, where previous studies did consider similar conditions, did the results agree?

Response: The effect of a polyelectrolyte layer on CO₂ reduction **in neutral electrolyte** has been studied. Alkali cations and HCO₃⁻ anions are the major ionic species in neutral electrolyte. These studies show that the polyelectrolyte can modulate the concentration distribution of HCO₃⁻, CO₃²⁻ and OH⁻ through Donnan exclusion and tune the pH in the polyelectrolyte layer. Besides, it was also reported that the polyelectrolyte layer can modulate the local CO₂ and H₂O concentration. Our research was focused on **acidic electrolyte**, in which H⁺ is the major ionic species. The polyelectrolyte also modulates the concentration distribution of H⁺ through Donnan exclusion. More importantly, we

investigated the effect of polyelectrolyte on the mass transport rate of H^+ , which is important to understand the rate of HER. We also investigated the effect on electric field distribution in double layer, which is important to understand the rate of CO_2 reduction. The latter two effects only involved in acidic electrolyte and thus were not discussed in the previous reports. In summary, the polyelectrolyte affects CO_2 reduction through different ways in acidic and neutral conditions. Therefore, the current work is innovative with respect to the studies in neutral condition.

In page 14, we added “In neutral electrolyte, the polyelectrolyte can modulate the concentration distribution of HCO_3^- , CO_3^{2-} and OH^- through Donnan exclusion and thus tune the local pH.^{41,42} It was also reported that the polyelectrolyte layer can affect the local water content and CO_2 concentration.^{24,42} In acidic electrolyte, the concentration distribution of H^+ can also be tuned by the polyelectrolyte through Donnan exclusion. More importantly, the polyelectrolyte layer may affect the rate of H^+ mass transport and hence determine the rate of H_2 evolution from H^+ reduction. The polyelectrolyte may also modulate the electric field in Stern layer and determines the rate of the electron transfer from cathode to CO_2 .”

Comment 2: From Figure 5c, I understand that the polymer layer is 1 micrometer thick. However, I overlooked this fact in the main manuscript. It would be wise to state this more clearly.

Response: Our simulation was based on our MDE experiments. In the MDE experiment, the thickness of the c-PDDA layer was around 1 μm according to the loading of c-PDDA on Ag MDE. Therefore, we set the thickness of polymer layer at 1 μm in our simulation.

We added the explanation above at page 16: “According to the loading of c-PDDA on Ag MDE in the experiments, the thickness of the polymer layer was set to 1 μm in the simulation.”

Comment 3: Looking at the different charge densities considered, I understand the Donnan potentials seen in figure 5c, but the rest of the profiles (zero or varying electric fields inside or beyond the polymeric layer) are not completely clear to me.

Response: We further depicted the profiles of electric field strength (toward the cathode) with polymer layer with different charge density, as shown in Figure S22. The profile of electric field strength is clearer to show the change of potential than the profile of potential itself. For charged polymer layers, ultra-strong field exists at the polymer-solution interface due to the potential step at the interface as shown in Figure 5c. For neutral polymer layer, the electric field strength varies in the region close to the cathode (black curve). For negatively charged polymer layer (magenta curve), the electric field strength varies in the solution close to the polymer. For all the other region, the curves of electric field strength are quite flat. It is noteworthy that the electric field strength in the solution decreases in the sequence of $\rho = -100 > \rho = 0 > \rho = +100 > \rho = +300$, in accordance with the sequence of migration rate of H^+ shown in Figure 5b.

In page 17, we added “Figure S22 shows the profiles of electric field strength with different ρ_p . Strong electric field exists at the interface between charged polymer layer and solution due to the potential step at the interface. For Ag electrode covered by cationic polymer layer, the electric field is uniform within the polymer layer and solution. It is noteworthy that the electric field strength outside the polymer layer decreases as ρ_p increases, in accordance with the migration rate of H^+ shown in Figure 5b.”

Figure S22. Profiles of electric field strength toward the cathode on Ag electrode covered by polymer layer with different ρ_p (unit: C·cm⁻³) in 10 mM HOTf. (a) Linear scale. (b) Logarithmic scale.

Comment 4: The electric field in the Stern layer is extracted from the GMPNP simulations, which aim to account for steric effect, but still might not be so accurate directly at the interface.

In the first place, the local properties in the EDL can be strongly affected by electrostatic correlations and molecular orientations. The authors account for a (rather drastic) drop of 90% in diffusion coefficient but for example not the effect of the polymer charge or local hydronium concentration on the dielectric permittivity. Could

the authors justify their choice? For reference: Zhu *et al.* (ref 46 in the manuscript) assumed a drop of over 90% in permittivity in the Stern layer. A correlation to relate the local permittivity to concentration (in the case of free ions) was used for example by Bohra *et al.* (ref 41), and in a recent article that builds on the work of Bohra *et al.*: Butt *et al.* *Sustainable Energy and Fuels* 7, 144-154 (2023), <https://doi.org/10.1039/D2SE01262F> (notably, that article also uses a Frumkin kinetic model in line with the present manuscript)

A change in the local permittivity would strongly affect the calculated E_{stern} and corresponding potential drop. As the authors mention in line 333, this is the driving force for the CO₂RR.

Response: We do agree that the relative permittivity (ϵ_r) is sensitive to the local environment. To verify whether the variation of diffusion coefficients (D_i) and ϵ_r affects the conclusion of the simulation, we changed D_i and ϵ_r in the polymer layer and conducted the GMPNP simulations. We set the diffusion coefficient of species i ($i = \text{H}^+$ and OTf⁻) in the polymer layer as:

$$D_{i,p} = x \cdot D_{i,s} \quad (\text{S26})$$

$D_{i,p}$ and $D_{i,s}$ are the diffusion coefficients of i in polymer layer and solution, respectively. We tried different values of x . We also tried to tune the ϵ_r in the polymer layer. Tables S2 and S3 summarize the simulation results with different D_i and different ϵ_r , respectively. The migration rate of H^+ and the electric field strength in Stern layer (E_{stern}) at -1.8 V vs SHE are summarized. The variation of $D_{i,p}/D_{i,s}$ ratio did not lead to significant change of E_{stern} . The migration rate of H^+ increased as the $D_{i,p}/D_{i,s}$ ratio decreased, but the trend that the migration rate of H^+ decreases as ρ_p increases did not change. Therefore, the variation of $D_{i,p}/D_{i,s}$ ratio does not affect the conclusion of the simulation.

Moreover, the decrease of $\epsilon_{r,p}$ did not lead to significant change of the migration rate of H^+ . E_{stern} increased as $\epsilon_{r,p}$ decreased, but the trend that E_{stern} increases as ρ_p increases did not change. Therefore, the variation of $\epsilon_{r,p}$ does not affect the conclusion

of the simulation.

Table S2. Simulated migration rate of H⁺ (J_{Mig}) and the electric field strength in Stern layer (E_{Stern}) at -1.8 V vs SHE on Ag electrode covered by polymer layer with different charge density (ρ_{p}). The ratio between diffusion coefficients in polymer layer and in solution ($D_{i,p}/D_{i,s}$) was set to 0.1 or 0.5 and the simulation results are compared.

ρ_{p} (C·cm ⁻³)	$D_{i,p}/D_{i,s} = 0.1$		$D_{i,p}/D_{i,s} = 0.5$	
	J_{Mig} ($\mu\text{mol}\cdot\text{cm}^{-2}\cdot\text{s}^{-1}$)	E_{Stern} (V·nm ⁻¹)	J_{Mig} ($\mu\text{mol}\cdot\text{cm}^{-2}\cdot\text{s}^{-1}$)	E_{Stern} (V·nm ⁻¹)
-100	1.10	0.343	1.10	0.343
0	0.56	0.387	0.91	0.407
+100	1.96×10^{-3}	0.550	9.64×10^{-3}	0.551
+300	2.60×10^{-4}	0.888	1.30×10^{-3}	0.888

Table S3. Simulated migration rate of H⁺ (J_{Mig}) and the electric field strength in Stern layer (E_{Stern}) at -1.8 V vs SHE on Ag electrode covered by polymer layer with different charge density (ρ_{p}). The relative permittivity of the polymer layer ($\epsilon_{\text{r,p}}$) was set to 80.1 or 50.0 and the simulation results are compared.

ρ_{p} (C·cm ⁻³)	$\epsilon_{\text{r,p}} = 80.1$		$\epsilon_{\text{r,p}} = 50.0$	
	J_{Mig} ($\mu\text{mol}\cdot\text{cm}^{-2}\cdot\text{s}^{-1}$)	E_{Stern} (V·nm ⁻¹)	J_{Mig} ($\mu\text{mol}\cdot\text{cm}^{-2}\cdot\text{s}^{-1}$)	E_{Stern} (V·nm ⁻¹)
-100	1.10	0.343	1.10	0.404
0	0.56	0.387	0.55	0.446
+100	1.96×10^{-3}	0.550	1.87×10^{-3}	0.676
+300	2.60×10^{-4}	0.888	2.48×10^{-4}	1.070

Considering that the orientation of water molecules and the chemisorption on the cathode lead to significant decrease of ϵ_r in Stern layer as illustrated by Zhu *et al.*, we modified the value of ϵ_r in Stern layer in Equation S19 to 10% of ϵ_r of water, namely to

8.01. Table S4 shows the simulation results with different values of ϵ_r in Stern layer ($\epsilon_{r,\text{Stern}}$). The decrease of $\epsilon_{r,\text{Stern}}$ did not lead to significant change of the migration rate of H^+ . E_{Stern} increased drastically as $\epsilon_{r,\text{Stern}}$ decreased, but the **difference** among the values of E_{Stern} at varied ρ_p kept almost unchanged as $\epsilon_{r,\text{Stern}}$ decreased. Considering that the electric field in Stern layer is the energetic driving force of CO_2 reduction, the decrease of $\epsilon_{r,\text{Stern}}$ would not lead to the change of the **relative rate** of CO_2 reduction obtained with different ρ_p . Therefore, the variation of $\epsilon_{r,\text{Stern}}$ does not affect the conclusion of the simulation.

Table S4. Simulated migration rate of H^+ (J_{Mig}) and the electric field strength in Stern layer (E_{Stern}) at -1.8 V vs SHE on Ag electrode covered by polymer layer with different charge density (ρ_p). The relative permittivity in Stern layer ($\epsilon_{r,\text{Stern}}$) was set to 80.1 or 8.01 and the simulation results are compared.

ρ_p ($\text{C}\cdot\text{cm}^{-3}$)	$\epsilon_{r,\text{Stern}} = 80.1$		$\epsilon_{r,\text{Stern}} = 8.01$	
	J_{Mig} ($\mu\text{mol}\cdot\text{cm}^{-2}\cdot\text{s}^{-1}$)	E_{Stern} ($\text{V}\cdot\text{nm}^{-1}$)	J_{Mig} ($\mu\text{mol}\cdot\text{cm}^{-2}\cdot\text{s}^{-1}$)	E_{Stern} ($\text{V}\cdot\text{nm}^{-1}$)
-100	1.10	0.343	1.07	2.012
0	0.56	0.387	0.50	2.051
+100	1.96×10^{-3}	0.550	1.95×10^{-3}	2.240
+300	2.60×10^{-4}	0.888	2.60×10^{-4}	2.548

We also tried to correlate the local ϵ_r with the local concentration of cation species in the GMPNP simulation according to Bohra's reports, but we failed to get convergent solution of the partial differential equations in our simulated case.

Above simulations with varied D_i and ϵ_r are added in the **Supplementary Note 2: GMPNP simulations with varied diffusion coefficients and relative permittivity**. In page 26, we added "**Considering that the values of D_i are affected by the structure of polymer and the value of ϵ_r varies according to the local environment, GMPNP simulations with different values of D_i and ϵ_r were also conducted to check whether the variation of these**

parameters affects the conclusion of the simulation. The results are summarized in Supplementary Note 2: GMPNP simulations with varied diffusion coefficients and relative permittivity.”

To Reviewer 3:

General Comments: This work achieved a significant enhancement of product selectivity and stability for the CO₂ electroreduction in an acidic catholyte free from metal cations by coating the catalyst with cross-linked diallyldimethylammonium (PDDA)-based polymer. Through a combined experimental and theoretical investigation, the authors reported that the high density of positively charged functional groups of the PDDA could serve a similar role of the alkali cations in retarding the proton migration close to the catalyst surface and enhancing the electric field within the Stern layer, and thus promote the electrode activity and stability in reducing CO₂ in acidic environment.

The loss of CO₂ in (bi)carbonates and salt precipitation are critical challenges limiting the application of CO₂ electrolysis at a scale. CO₂ electrolysis in an acidic environment free from metal cations is promising route in addressing these challenges. Therefore, the findings from this work are timely and interesting. However, there are a few concerns listed below for the authors to consider.

Response: We highly appreciate the reviewer's comments. The point-to-point response to the comments can be found below. The corresponding revisions in the **main text** and **Supplementary Information** are highlighted in **yellow**.

Comment 1: Recent reports (e.g., Nature Catalysis, 2021, 4, 654-662) highlighted that the metal cations are essential in activating CO₂ reduction via stabilizing the CO₂-intermediate via a short-range electrostatic interaction. The results from this work indicated that the non-metal cationic groups could also activate CO₂ reduction, which can be an interesting alternative perspective to the current understanding. However, it remains unclear to me in the main text how CO₂ reduction could proceed within the polymer environment. If the cationic site behaves similarly to a metal cation that stabilizes the intermediate via a short-range interaction, will the steric hinderance be an

issue for the polymer? It would be also good if the authors could experimentally examine the polymer properties before and after high-rate CO₂ electrolysis.

Response: According to our GMPNP simulation with the continuum electrolyte model, both cationic polymer layer and K⁺ lead to increase of the electric field strength in Stern layer, which can stabilize the polar *CO₂ intermediate. The explicit short-range electrostatic interaction between K⁺ or quaternary ammonium cation and *CO₂ may augment the stabilization energy of *CO₂, but this effect is not considered in our simulation. In the recent report (ref. 29: *Nat. Catal.* **2021**, 4, 654-662), the interaction between partially dehydrated alkali cation at OHP and *CO₂ was considered indispensable for triggering CO₂ reduction. However, the recent work of Zhuang *et al.* realized CO₂ reduction on MEA with pure water as the anolyte. In their reports (ref. 39: *Nat. Energy* **2022**, 7, 835; ref. 40: *Electrochim. Acta* **2023**, 458, 142509), the catalyst was coated with an ionomer with quaternary ammonium as the immobilized cationic site, which enabled CO₂ reduction in a condition free of alkali cations. Therefore, we hypothesize that the quaternary ammonium cations may also have short-range interaction with *CO₂ species and promote CO₂ reduction. Steric hinderance should be a great issue for the interaction between N-based cations and adsorbed species. For instance, Koshy *et al.* reported that the interaction between a functionalized imidazolium cation and an adsorbed bicarbonate species weakens as the substituent group on the imidazolium becomes bulkier (ref. 41: *JACS* **2021**, 143, 14712). The quaternary ammonium site on PDDA bears two methyl groups, the smallest substituent group. Therefore, the quaternary ammonium site on PDDA should show stronger interaction with *CO₂ species than the quaternary ammonium site with other larger substituent groups. Since the quaternary ammonium site cannot directly bind to *CO₂, this kind of interaction should be weaker than that between alkali cation and *CO₂. Taking all of these effects into account, c-PDDA should exert weaker promotion effect on CO₂ reduction than K⁺ cations. This is confirmed by the experimental result in Figure S19: The applied overpotential to reach the same partial current density of CO₂ reduction on c-PDDA decorated catalysts in 0.1 M H₂SO₄ is larger than that on bare

catalysts in 0.1 M H₂SO₄ + 0.4 M K₂SO₄. Through our strategy, larger overpotential is the price paid for the improved stability. In our opinion, for a sustainable technique, better stability is more important than smaller overpotential.

Figure S19. Plots of partial current density of CO dependent on the potential of working electrode for c-PDDA decorated Ag NPs in 0.1 M H₂SO₄ and bare Ag NPs in 0.1 M H₂SO₄ + 0.4 M K₂SO₄.

The discussion on this aspect was added in page 13: “It was reported that immobilized quaternary ammonium cations on ionomer can enable CO₂ reduction on membrane electrode assembly (MEA) with pure water as the anolyte,^{39,40} implying that the quaternary ammonium cations have the ability to interact with *CO₂ species and promote CO₂ reduction in alkali cation-free condition. The interaction between quaternary ammonium cation and the adsorbed species weakens as the substituent groups on the N atom become bulkier.⁴¹ The N atoms in PDDA bears two methyl groups, the smallest substituent group. Therefore, PDDA should show stronger interaction with *CO₂ species than quaternary ammonium cations with other substituent groups. In the alkali cation-containing electrolyte, the partially dehydrated alkali cation at OHP can bind to *CO₂ species, which is essential for triggering CO₂ reduction.^{29,33} Since the quaternary ammonium cation cannot directly bind to *CO₂, the short-range interaction between *CO₂ and the quaternary ammonium cation should be weaker than that between *CO₂ and alkali cations. In addition, both alkali cations and quaternary ammonium cations can increase the electric field strength in Stern layer, which also stabilizes the polar *CO₂ intermediate, as discussed in the following section. Taking all

the effects into account, K^+ should show more profound promotion effect on the kinetics of CO_2 reduction than c-PDDA, in accordance with our observation that the applied potential to reach the same partial current density of CO on c-PDDA decorated Ag NPs in K^+ -free electrolyte was more negative than on bare Ag NPs in K^+ -containing electrolyte (Figure S19).”

To examine the property of the polymer layer before and after electrolysis, electrochemical impedance spectra (EIS, Figure S8) and infrared spectra (IR, Figure S7) of the working electrode were measured. No substantial change in the spectra was observed, indicating the Ag-polymer interface and the chemical nature of the polymer were stable during electrolysis. In page 9, we added: “Figure S7 and S8 shows the infrared (IR) spectra and the electrochemical impedance spectra (EIS) of the working electrode before and after electrolysis, respectively. No substantial change in the spectra was observed, indicating both the polymer layer and the interface between Ag NPs and the polymer were stable during electrolysis.”

Figure S7. IR spectra of c-PDDA decorated Ag NPs on GDE before and after CO_2 reduction experiment. The stretching vibration of C-H and C-N bonds, and the bending vibration of C-H bonds are assigned.

Figure S8. (a) EIS spectra and (b) the fitting circuit of c-PDDA decorated Ag NPs on GDE in 0.1 M H₂SO₄ before and after CO₂ reduction experiment. The central potential was -1.0 V vs SHE. The hollow circles and solid lines in panel (a) are experimental data and fitting curves, respectively. In the fitting circuit, R_s , R_p and R_{ct} represent the resistance of solution, polymer layer and charge transfer at the surface of catalyst, respectively. C_p and C_{dl} are the capacitance of polymer layer and the electric double layer, respectively. Z_w is the Warberg impedance. The fitting values of R_s , R_p and R_{ct} are shown.

Comment 2: Following up the above question, the use of XPS results may not be sufficient to support the authors' claim that the PDDA was washed away in line 126 – 133. Can the PDDA be detected and quantified in the electrolyte after the test? This additional result could help further support the authors' explanation and rule out the potential chemical degradation of the PDDA under CO₂ reduction conditions.

Response: We quantified PDDA in the electrolyte after electrolysis by ¹H-NMR. The catalyst was PDDA decorated Ag NPs and the electrolyte was 0.1 M H₂SO₄. After

electrolysis, the electrolyte was neutralized by KOH and water was removed by evaporation. The residue was then dissolved by 400 μL of D_2O as the sample for NMR analysis. For the standard reference, PDDA solution containing the same amount of PDDA as that added on the working electrode was distilled to remove the water and then dissolved by 400 μL of D_2O . Figure S2 compares the ^1H -NMR spectra of PDDA dissolved from the PDDA/Ag catalyst (red curve) and the standard reference sample (black curve). The signal with the chemical shift between 1.0 and 1.8 ppm is assigned to the CH and CH_2 moieties unbound to nitrogen atom (red H atoms in the inset), which was used to quantify the amount of PDDA. About 90% of PDDA was washed into the electrolyte. We added in page 7: “ ^1H -NMR spectrum of the electrolyte after electrolysis (Figure S2) indicates that about 90% of PDDA was washed into the electrolyte.”

Figure S2. ^1H -NMR spectra of PDDA in the electrolyte after CO_2 reduction experiment (red curve) and the standard reference solution of PDDA (black curve). The catalyst was PDDA decorated Ag NPs and the electrolyte was 0.1 M H_2SO_4 . After electrolysis, the electrolyte was neutralized by KOH and water was removed by evaporation. The residue was then dissolved by 400 μL of D_2O as the sample for NMR analysis. For the standard reference, PDDA solution containing the same amount of PDDA as that added on the working electrode was distilled to remove the water and then dissolved by 400 μL of D_2O . The signal with the chemical shift between 1.0 and 1.8 ppm is assigned to the CH and CH_2 moieties unbound to nitrogen atom (red H atoms in the inset), which was used to quantify the amount of PDDA dissolved by the electrolyte.

Comment 3: In Figure 2c, why the pH of the H₂SO₄-K₂SO₄ catholyte increases so significantly while anolyte's pH remains stable across the test? The authors should elaborate more experimental details to justify this point. If the pH changed so drastically in the catholyte, I don't think it is a fair comparison of the CO Faradaic efficiencies particularly between H₂SO₄-K₂SO₄ and c-PDDA test shown in Figure 2a.

Response: According to the scheme in Figure 1b, the decreasing of the amount of H⁺ in the catholyte should equal to the increasing of the amount of H⁺ in the anolyte (buffering reaction is not considered). If most of H⁺ in catholyte is consumed while the concentration of H⁺ in the anolyte doubles, the pH of catholyte will increase drastically while the pH of anolyte will only decrease for about 0.3 unit ($\lg 2 = 0.301$). Therefore, our observation of the change of pH of catholyte and anolyte is reasonable. Because of the buffering effect of CO₂/HCO₃⁻, the pH of the catholyte was convergent to about 7.

Experimental details of CO₂ reduction in the flow cell were added into the Methods section, including the volumes of the catholyte and anolyte, flow rate of electrolyte and CO₂ gas: “The catholyte and the anolyte were circulated separately by two peristaltic pumps. The volumes of catholyte and anolyte were both 30 mL and the flow rates were both 10 mL·min⁻¹. CO₂ was fed through a gas chamber behind the GDE. The flow rate was fixed at 30 standard cubic centimeter per minute (sccm) by a mass flow controller.”

To verify that the decrease of FE of CO in H₂SO₄-K₂SO₄ is not due to the increase of pH of the catholyte, we measured the initial FE of CO in electrolyte with different pH. The electrolyte contained x M H₂SO₄ + 0.4 M K₂SO₄ ($x = 0, 0.001, 0.1$). As shown in Figure S11, all the initial FE of CO is around than 90%. Therefore, the drastic increase of catholyte pH is not the direct reason for the decrease of FE of CO in H₂SO₄-K₂SO₄. The decrease of FE of CO was caused by the formation of bicarbonate precipitate. The FE of CO on c-PDDA decorated Ag electrode in 0.1 M H₂SO₄ was stable since no bicarbonate precipitate was formed. Therefore, we think the comparison of the FE of CO between H₂SO₄-K₂SO₄ and c-PDDA is fair. In page 11, we added: “It

is noteworthy that the increase of the pH of the catholyte did not directly cause the decrease of the FE of CO on bare Ag NPs. As shown in Figure S11, the initial FEs of CO on bare Ag NPs in electrolytes with varied pH are all around 90%. The decrease of the FE of CO in 0.1 M H₂SO₄ + 0.4 M K₂SO₄ in Figure 2a was a direct consequence of the formation of KHCO₃ precipitate.”

Figure S11. FEs of CO on bare Ag NPs in K⁺-containing electrolyte with different pH. The electrolyte contained x M H₂SO₄ + 0.4 M K₂SO₄ ($x = 0, 0.001, 0.1$). The current density was $-200 \text{ mA}\cdot\text{cm}^{-2}$. The FE was measured at 10 minutes of electrolysis.

Comment 4: The authors stated in line 192 – 193 that the loss of the electrode hydrophobicity is a result of the salt precipitation. However, there is no solid evidence from this work to support this statement. In addition, the polymer coated on the catalyst seems to be more hydrophobic than the bare catalyst based on metallic nanoparticles. The limited electrode flooding can be also partially contributed by the hydrophobic polymer coating. The authors should address this point in the main text.

Response: We tested the contact angle of the working electrodes with and without c-PDDA to characterize the hydrophobicity, as shown in Figure S10. Typically, the side with catalyst is more hydrophilic to ensure the sufficient contact between catalyst and electrolyte. Due to the high cation density, c-PDDA is very hydrophilic, not

hydrophobic. The contact angle on Ag NPs decorated by c-PDDA (59°) is drastically smaller than on bare Ag NPs (152°), as shown by Figure S10d-e. The back side of GDE (the side of gas diffusion layer) is typically more hydrophobic to ensure the smooth mass transport of CO_2 gas. The EDS mapping in Figure S9 indicates that the KHCO_3 precipitate aggregate at the side of gas diffusion layer. The contact angle at the back side of GDE after electrolysis in $\text{H}_2\text{SO}_4\text{-K}_2\text{SO}_4$ (141°) was considerably lower than that after electrolysis in H_2SO_4 (161°), indicating the formation of KHCO_3 precipitate reduced the hydrophobicity of the gas diffusion layer of GDE. This would lead to flooding and blocking of the mass transport of CO_2 , which is the direct cause of the decrease of FE of CO in $\text{H}_2\text{SO}_4\text{-K}_2\text{SO}_4$.

In page 10, we added: “As shown in Figure S10, the contact angle of water on the side of gas diffusion layer of the working electrode after electrolysis in 0.1 M H_2SO_4 + 0.4 M K_2SO_4 was considerably smaller than that after electrolysis in 0.1 M H_2SO_4 , indicating that the formation of KHCO_3 precipitate reduced the hydrophobicity of the GDE.”

Figure S10. Contact angles of GDEs with bare Ag NPs and c-PDDA decorated Ag NPs. (a) Back side (gas diffusion layer side) of GDE before CO_2 reduction. (b) Back side of c-PDDA/Ag/GDE after CO_2 reduction in 0.1 M H_2SO_4 . (c) Back side of bare Ag/GDE after CO_2 reduction in 0.1 M H_2SO_4 + 0.4 M K_2SO_4 . (d) Catalyst side of bare Ag/GDE before CO_2 reduction. (e) Catalyst side of c-PDDA-Ag /GDE before CO_2 reduction.

Comment 5: When experimentally examining the effect of the polymer adlayer on

proton migration and interfacial electric field, the authors used HOTf as the supporting electrolyte, which is different from the reaction environment for CO₂ electroreduction. The authors should explain why these two sets of experiment are translatable and there is no potential impact from the trifloromethanesulfonate anions. I also found the explanation in line 239 – 244 lacks solid supporting evidence.

Response: HOTf is a strong acid in water while the second proton of H₂SO₄ is not a strong acid. Our GMPNP modeling was designed to simulate the MDE experiment. To simplify the model, we chose HOTf as the electrolyte. For a useful and low-cost CO₂ reduction technique, H₂SO₄ is a better choice for the electrolyte. Another possible choice of the electrolyte is HClO₄. However, ClO₄⁻ may be reduced to generate trace amount of Cl⁻ anions which can be specifically adsorbed on Ag. In contrast, OTf⁻ is unlikely to be specifically adsorbed on Ag.

In page 14, we added: “HOTf instead of H₂SO₄ was used as the electrolyte for the MDE experiments since HOTf dissociates completely in water, which helps to simplify the GMPNP modeling.”

Comment 6: In the model, is it reasonable to assume the charge density of the polymer is uniform across the polymer layer? If not, how will the cationic sites be distributed in the polymer layer, and will such distribution changes the conclusion from the modelling results? The authors should talk about the limitation of their models.

Response: Since the cationic sites on c-PDDA is immobilized on the backbones of the polymer, the charge density of the polymer should not vary significantly across the polymer layer. If the polymer can be compressed by the electrostatic attracting force generated from the cathode, the density of cationic site near the cathode may increase slightly. To check the influence of the non-uniformity of the charge density on the modeling result, we simulated a polymer layer with higher charge density on the cathode side and lower charge density on the solution side, as shown in Figure S25. Compared with a polymer layer with identical total charge and uniform charge density,

the non-uniform charge density leads to decrease of the migration rate of H^+ and increase of the electric field strength in Stern layer. Both effects can lead to the improved selectivity of CO_2 reduction.

In page 21, we added: “If the polymer layer can be compressed under the electrostatic attraction generated from the cathode, ρ_p at the cathode side should increase. As shown in Figure S25, the accumulation of cationic site to the cathode side leads to lower migration rate of H^+ and higher E_{Stern} . Both effects result in improved selectivity of CO_2 reduction.”

Figure S25. Simulated effects of the uniformity of the polymer layer on the properties of Ag electrode in 10 mM HOTf. (a) The proposed profiles of ρ_p for polymer with uniform charge density and non-uniform charge density. (b) The migration rate of H^+ at 2 μm from the OHP. (c) Plots of the electric field strength in Stern layer based on the electrode potential.

Comment 7: Figure 5d shows that the polymer with 300 positive charge $C \text{ cm}^{-3}$ did not show a local pH as high as the case with K^+ . Is a $\text{pH} = 3$ sufficient to limit the availability of protons for the competitive HER? When calculating the local pH in the model, did the authors consider the water content within the polymer? The polymer with more fixed charges is expected to have more water molecules, does the model capture this feature?

Response: In this GMPNP simulation, only H^+ reduction was considered. CO_2 reduction was not considered. According to the report of Koper *et al.* (ref. 12: *JACS* **2021**, *143*, 279), in a mildly acidic electrolyte ($\text{pH} \approx 3$), OH^- generated from CO_2 reduction can neutralize H^+ and suppress H^+ reduction. If we consider OH^- anions generated from CO_2 reduction, the local pH should be higher than 3 and H^+ reduction can be further suppressed. Since CO_2 reduction was conducted on GDE and GMPNP modeling of GDE is quite difficult, we did not consider this effect in our simulation. This explanation was added in page 19: “When CO_2 reduction is involved, OH^- anions generated from CO_2 reduction can neutralize H^+ and lead to further increase of local pH. Therefore, the local pH under CO_2 reduction condition should be higher than the value shown in Figure 5d.”

In our GMPNP simulation, the water content in a polymer layer is reflected by the volume fraction of aqueous solution in the polymer layer ($1 - V_p$). V_p is the volume fraction of polymer. This parameter affects the activity of H^+ according to Equation S3 and thus affects the simulated pH value.

In our original simulation, V_p was set to 0.36 according to the experimental swelling ratio of c-PDDA. For the simulation of polymer layer with different ρ_p , we used the same value of V_p in our previous manuscript. According to the reviewer’s comment, as ρ_p decreases, the value of $(1 - V_p)$ should decrease, corresponding to less solution accommodated in the polymer layer. Therefore, in this manuscript, we set V_p to 0.68 for the simulation of polymer layers with ρ_p smaller than $300 \text{ C} \cdot \text{cm}^{-3}$. Thus, the

volume fraction of aqueous solution in the polymer layer decreases to half of that in the polymer layer with $\rho_p = +300 \text{ C}\cdot\text{cm}^{-3}$. Please see page 3 of the Supplementary Information for the revision. The change of the value of V_p did not affect the conclusion of the simulation that the migration rate of H^+ decreases as ρ_p increases (Figure 5b), local pH at OHP increases as ρ_p increases (Figure 5d) and electric field strength in Stern layer increase as ρ_p increases (Figure 6b).

Comment 8: The authors should include the error bars calculated from three repetitive experiments for their key results. It is necessary to help the field to understand the repeatability of the results.

Response: The measurements of mass of electrolyte permeating the GDE (Figure 2d), and the CO_2 reduction performances of c-PDDA decorated Ag NPs and In NPs (Figure 3a, b) were repeated for 3 times. The error bars were added in these figures.

Figure 2. ... (d) Mass of electrolyte permeating through the cathode after electrolysis

with constant current density of $-200 \text{ mA}\cdot\text{cm}^{-2}$ for 10 hours. Error bars are the standard deviations based on three individual measurements.

Figure 3. CO₂ reduction performances of c-PDDA decorated catalysts in 0.1 M H₂SO₄. (a) Ag NPs and (b) In NPs were used as the catalysts. Chronopotentiometry experiments were conducted. The FEs of H₂ (grey), CO (orange) and formic acid (blue), and the electrode potential (dark blue curves) are shown. Error bars are the standard deviations based on three individual measurements.

Comment 9: The experimental section, I could not see whether or not the authors measured the outlet flow rate from the flow cell. It is important to measure the outlet flow rates to calculate the FEs and understand the carbon utilization efficiency. Additionally, the authors mentioned in the experimental section that they controlled the gas by using a mass flow meter, which guess should be a mass flow controller. A meter can only measure the flow rate.

Response: The flow rate of the outlet of the flow cell was measured by a soap film flowmeter and the FEs of gas phase products were calculated based on this flow rate. In page 25, we added this experimental detail: “The FEs of gas phase products were calculated based on the flow rate of the outlet gas from the flow cell measured by a soap film flowmeter.” The inlet flow rate of CO₂ was controlled by a mass flow controller. We revised ‘mass flow meter’ to ‘mass flow controller’.

To Reviewer 4:

General comments: CO₂ reduction in acidic condition is a good idea for avoiding the formation of carbonate during CO₂ reduction. The idea of using cross-linked poly(2,2'-diallyldimethylammonium chloride) in the system of free cation is very interesting and the results of stability and selectivity are very encouraging. This manuscript requires a major revision before publication on Nature Communication. I have few questions and comments as below.

Response: We highly appreciate the reviewer's comments. The point-to-point response to the comments can be found below. The corresponding revisions in the **main text** and **Supplementary Information** are highlighted in **yellow**.

Comment 1: What is the potential of oxidation reaction in anodic compartment? And the author should show and discuss about the cell potential during electrolysis. It is important to see the advantage as well as disadvantage of the system for CO₂ reduction in acidic condition.

Response: Figure S18 shows **overall cell potential** and the components of potential loss at varied current density, including Ohmic loss, cathode overpotential (for CO₂ reduction) and **anode overpotential (for oxygen evolution reaction)**. 0.1 M H₂SO₄, 0.1 M KHCO₃ and 0.1 M KOH were used as acidic, near neutral and alkaline electrolytes, respectively. Since a flow cell with the cathode-anode distance of 5 mm was used throughout our experiments, the major potential loss at high current density was the Ohmic loss. The cell potential with 0.1 M H₂SO₄ was the smallest since the acidic electrolyte shows the lowest resistance. The cell potential with 0.1 M KHCO₃ is the largest due to the highest resistance of the near neutral electrolyte. For 0.1 M KOH, the apparent cathode overpotential is the lowest. However, due to the reaction between KOH and CO₂, KOH solution is not sustainable during electrolysis and is not a practical choice of the electrolyte for CO₂ reduction techniques.

In page 12, we added the comparison among different electrolytes: “Figure S18 further compares the overall cell potential and each component of the potential loss at different current densities with 0.1 M H₂SO₄, 0.1 M KHCO₃ and 0.1 M KOH as the electrolyte. The potential loss is composed of ohmic loss, overpotentials of cathodic reaction and anodic reaction. The cell potential with 0.1 M H₂SO₄ is the lowest due to the lowest resistance of the electrolyte. The apparent overpotential of CO₂ reduction in 0.1 M KOH is the lowest,^{9,25} but KOH solution is not sustainable during electrolysis and is not a practical choice as the electrolyte for CO₂ reduction techniques. The cell potential with 0.1 M KHCO₃ is the largest due to the highest resistance of the electrolyte.”

Figure S18. Comparison of the overall cell potential and the potential losses due to electrolyte resistance (Ohmic loss), cathode overpotential (η of CO₂RR) and anode overpotential (η of OER) at varied current densities with different electrolytes. 0.1 M H₂SO₄, 0.1 M KHCO₃ and 0.1 M KOH were used as the electrolytes. The cathode was c-PDDA decorated Ag NPs on GDE. The anode used in acidic and near neutral electrolytes was an IrO₂-decorated Ti foil, and the anode used in alkaline electrolyte was an Fe-decorated Ni foam.⁹

Comment 2: I would recommend the author showing the result of electrochemical impedance before and after electrolysis of CO₂ reduction. It is important to understand more about electrochemical properties of cathodic electrode during CO₂ reduction.

Response: Electrochemical impedance spectra (EIS) of c-PDDA decorated Ag NPs on

GDE before and after CO₂ reduction were measured, as shown in Figure S8a. The spectra before and after CO₂ reduction electrolysis are similar. The spectra were fitted based on the circuit depicted in Figure S8b. For the surface of catalyst, the resistance of charge transfer (R_{ct}), Warberg impedance (Z_w) and capacitance of double layer (C_{dl}) are considered. The resistance and capacitance of the polymer layer (R_p and C_p) are considered. The resistance of solution (R_s) is also involved. The values of R_s , R_p and R_{ct} before and after CO₂ reduction are shown in Figure S8b. These values did not show significant change after electrolysis, indicating the Ag-polymer interface was stable during electrolysis.

Figure S8. (a) EIS spectra data (circles) and fitting curves (solid lines) and (b) the fitting circuit of c-PDDA decorated Ag NPs on GDE in 0.1 M H₂SO₄ before and after CO₂ reduction experiment. The central potential was -1.0 V vs SHE. The hollow circles and solid lines in panel (a) are experimental data and fitting curves, respectively. In the fitting circuit, R_s , R_p and R_{ct} represent the resistance of solution, polymer layer and charge transfer at the surface of catalyst, respectively. C_p and C_{dl} are the capacitance of polymer layer and the electric double layer, respectively. Z_w is the Warberg impedance. The fitting values of R_s , R_p and R_{ct} are shown.

REVIEWERS' COMMENTS

Reviewer #1 (Remarks to the Author):

The authors answered and addressed the reviewer's questions and concerns in the updated manuscript.

Reviewer #2 (Remarks to the Author):

The authors clarified all my points and also addressed the insightful points from the other reviewers in detail. I now support the publication of this manuscript.

I did wonder about one minor thing, though. Figure S19 shows a comparison of the partial current density of CO dependent on the potential of the working electrode for the case of alkali cations and polyelectrolyte. The result of such a comparison would depend on the amount of polyelectrolyte (surface coating is controlled) and alkali solutions (bulk concentration is controlled). It is not clear to me to what extent the authors present a fair comparison.

Reviewer #3 (Remarks to the Author):

I am satisfied with the authors' response to my queries and the corresponding revision.

To Reviewer #1:

Comment: The authors answered and addressed the reviewer's questions and concerns in the updated manuscript.

Response: We thank the reviewer for the comment and highly appreciate the recognition of our efforts.

To Reviewer #2:

Comment: The authors clarified all my points and also addressed the insightful points from the other reviewers in detail. I now support the publication of this manuscript.

I did wonder about one minor thing, though. Figure S19 shows a comparison of the partial current density of CO dependent on the potential of the working electrode for the case of alkali cations and polyelectrolyte. The result of such a comparison would depend on the amount of polyelectrolyte (surface coating is controlled) and alkali solutions (bulk concentration is controlled). It is not clear to me to what extent the authors present a fair comparison.

Response: We thank the reviewer for the comment and highly appreciate the recognition of our efforts.

In Figure S19 of the present manuscript, we added the j_{CO} -potential plots of bare Ag catalyst in acidic electrolyte with varied concentration of K^+ . The overpotential to reach the same partial current density of CO on increases as the concentration of K^+ decreases, while the overpotential in 0.1 M H_2SO_4 + 0.04 M K_2SO_4 is still lower than that of c-PDDA decorated Ag catalyst in 0.1 M H_2SO_4 . The concentration of K^+ can be tuned while the concentration of cationic sites in the c-PDDA layer is constant. Therefore, we gave the range of concentration of K^+ in page 10 of the main text: “the applied potential to reach the same partial current density of CO on bare Ag NPs in K^+ -containing electrolyte (with K^+ concentration of 0.08~0.8 M) was more positive than on c-PDDA decorated Ag NPs in K^+ -free electrolyte (Figure S19).”

Figure S19. Plots of partial current density of CO dependent on the potential of working electrode for c-PDDA decorated Ag NPs in 0.1 M H₂SO₄ and bare Ag NPs in 0.1 M H₂SO₄ + 0.4 M K₂SO₄, 0.1 M H₂SO₄ + 0.1 M K₂SO₄ and 0.1 M H₂SO₄ + 0.04 M K₂SO₄.

To Reviewer #3:

Comment: I am satisfied with the authors' response to my queries and the corresponding revision.

Response: We thank the reviewer for the comment and highly appreciate the recognition of our efforts.